**RESEARCH**

# Single-cell multi-omics profiling links dynamic DNA methylation to cell fate decisions during mouse early organogenesis

Stephen J. Clark[1,2*†], Ricard Argelaguet[1,2*†], Tim Lohoff[1,3†], Felix Krueger[2,4], Deborah Drage[1,2], Berthold Göttgens[3,5], John C. Marioni[6,7,8], Jennifer Nichols[3,9,10] and Wolf Reik[1,3,8,2,11*]

†Stephen J. Clark, Ricard Argelaguet and Tim Lohoff contributed equally to this work.

*Correspondence:
sclark@altoslabs.com;
rargelaguet@altoslabs.com;
wreik@altoslabs.com

[2] Altos Labs Cambridge Institute of Science, Granta Park, Cambridge, UK
[11] Centre for Trophoblast Research, University of Cambridge, Cambridge, UK
Full list of author information is available at the end of the article

## Abstract

**Background:** Perturbation of DNA methyltransferases (DNMTs) and of the active DNA demethylation pathway via ten-eleven translocation (TET) methylcytosine dioxygenases results in severe developmental defects and embryonic lethality. Dynamic control of DNA methylation is therefore vital for embryogenesis, yet the underlying mechanisms remain poorly understood.

**Results:** Here we report a single-cell transcriptomic atlas from Dnmt and Tet mutant mouse embryos during early organogenesis. We show that both the maintenance and de novo methyltransferase enzymes are dispensable for the formation of all major cell types at E8.5. However, DNA methyltransferases are required for silencing of prior or alternative cell fates such as pluripotency and extraembryonic programmes. Deletion of all three TET enzymes produces substantial lineage biases, in particular, a failure to generate primitive erythrocytes. Single-cell multi-omics profiling moreover reveals that this is linked to a failure to demethylate distal regulatory elements in *Tet* triple-knockout embryos.

**Conclusions:** This study provides a detailed analysis of the effects of perturbing DNA methylation on mouse organogenesis at a whole organism scale and affords new insights into the regulatory mechanisms of cell fate decisions.

## Background

Early mammalian development is accompanied by epigenetic reprogramming. In the first phase of reprogramming, the DNA methylation (DNAm) marks of terminally differentiated germ cells are rapidly erased following fertilisation to produce the hypomethylated genome of the early embryo which is required for totipotency [1, 2]. Subsequently, the genome of the embryo proper undergoes de novo methylation such that high levels of CpG methylation are re-established shortly after implantation at embryonic day (E) 5.5 [2–5]. This hypermethylated state is maintained in

somatic cells, although the precise distributions of CpG methylation become highly tissue-specific indicating a role in cellular identity [6, 7]. It has been suggested that this global gain in DNA methylation, and the accompanying chromatin remodelling, is required to restrict the developmental potential and epigenetically prime cells for differentiation [8, 9].

DNA methylation is deposited by the de novo methyltransferases, DNMT3A and DNMT3B [10]. Once established, DNA methylation profiles are inherited through cell division by the activity of the maintenance methyltransferase, DNMT1 [11]. Embryos lacking *Dnmt1* do not survive past E9.5, are developmentally delayed and display a number of phenotypes including neural tube defects and lack of somites [12]. *Dnmt3a$^{-/-}$* mice develop to term, but are small in size and die around 4 weeks of age whereas deletion of *Dnmt3b* results in embryonic lethality at around E14.5 [10]. Double knockout of the two *de novo* methylases results in a similar phenotype to *Dnmt1* [10].

Removal of CpG methylation from the genome can be achieved by passive dilution, in which DNMT1 is prevented from copying methylation onto daughter strands during replication and this is the major contributor to global demethylation events [13]. Demethylation can also occur via enzymatic oxidation of methyl-cytosine into hydroxymethyl-cytosine and other oxidised derivatives catalysed by the ten-eleven-translocation (TET) family of enzymes [14–16]. These oxidised bases can be removed and replaced by unmodified cytosine by base excision repair [14, 17, 18] or can lead to replicative dilution due to UHRF1 evasion [19, 20].

Deletion of all three TET enzymes in mouse embryos leads to impaired growth at E7.5, primitive streak patterning defects, impaired maturation of mesoderm tissues and at E8.5, a failure to form the head fold, heart tissue, somites and gut tube [21].

Bulk RNA sequencing (RNA-seq) and bisulfite sequencing (BS-seq) of DNMT mutant embryos have provided insights into the role of DNAm in the repression of transposable elements, imprints and germline genes as well as zygotic and some lineage-specific genes [4, 22]. The remethylation of the genome which takes place between E4.5 and E6.5 in wildtype embryos is inhibited in both *Dnmt1* and *Dnmt3a/b* double knockout embryos, yet these survive as far as E9.5 [22]. Interestingly this suggests that DNAm is not essential for the first sets of lineage decisions, and only becomes deleterious after germ layer formation. However, the precise genomic elements responsible and the cell types affected are still unknown. Bulk RNA-seq and BS-seq of *Tet*-TKO embryos revealed mis-regulation of *Lefty1* and *Lefty2* and associated hypermethylation of nearby regulatory regions [21] but the precise cell type effects could not be revealed by this analysis.

Our current understanding of the effects of DNA methylation perturbations in embryogenesis is informed by morphological descriptions, immunofluorescence imaging and a limited amount of genome-wide analyses using bulk RNA-seq and BS-seq. Whilst informative, these studies are limited to the analysis of a small set of genes or lack of the ability to resolve cell type-specific effects. Single-cell RNA sequencing (scRNA-seq) of mutant embryos can address these limitations by providing a readout of cell type proportions together with an assessment of the cell type-specific molecular defects, as was recently demonstrated in a study that perturbed a number of epigenetic modifiers using CRISPR/Cas9 at the zygote stage [23]. In addition, single-cell multi-omics techniques such as scNMT-seq [3, 24], which profiles gene expression, DNA methylation and chromatin accessibility in single cells, can

provide additional information on the underlying epigenetic mechanisms of any defects observed.

To further investigate the perturbation of DNA methylation in this study we use scRNA-seq and targeted scNMT-seq to profile E8.5 embryos, representing the onset of organogenesis, in which *Dnmt1*, *Dnmt3a*, *Dnmt3b* and *Tet 1/2/3* have been disrupted.

## Results

### scRNA-seq of Dnmt3a[-/-], Dnmt3b[-/-] and Dnmt1[-/-] mutant embryos during mouse early organogenesis

We generated *Dnmt1*[-/-], *Dnmt3a*[-/-] and *Dnmt3b*[-/-] embryos together with matching wildtypes from heterozygous matings. We collected embryos at E8.5, when progenitor cells for all major organs have formed and methylation mutants are not yet lethal and performed scRNA-seq. To increase the statistical power of our analysis we combined our data set of KO embryos with a published data set where *Dnmt1*, *Dnmt3a* and *Dnmt3b* were disrupted using zygotic CRISPR-Cas9 injection and also profiled using scRNA-seq at E8.5 [23]. In total, our analysis comprises 51,811 cells from 17 WT embryos, 45,579 cells from 14 *Dnmt3a*[-/-] embryos, 55,237 cells from 12 *Dnmt3b*[-/-] embryos and 25,185 cells from 15 *Dnmt1*[-/-] embryos (Fig. 1a, Additional file 1: Fig. S1). We assigned cell type labels by mapping the RNA expression profiles to a comprehensive reference atlas that spans E6.5 to E8.5 [25] (Fig. 1b, c and Additional file 1: Fig. S2).

First, we assessed global cell fate defects by comparing the cell type proportions between KO and WT embryos (Fig. 1d, Additional file 1: Fig. S2). *Dnmt3a*[-/-] and *Dnmt3b*[-/-] embryos show relatively minor defects in cell type proportions, consistent with previous reports that indicate that these embryos do not display major defects during gastrulation [10]. In contrast, *Dnmt1*[-/-] embryos show widespread defects in cell type proportions, including a relative overrepresentation of extraembryonic (ExE) ectoderm (trophoblast) and immature embryonic cell types such as rostral neuroectoderm and caudal epiblast. We also observe a relative underrepresentation of some mature embryonic cell types, including neural crest, neuromesodermal progenitors (NMPs), brain, spinal cord and gut cells. The overrepresentation of ExE ectoderm is consistent with previous studies that found that Embryonic Stem

(See figure on next page.)

**Fig. 1** scRNA-seq of Dnmt3a[-/-], Dnmt3b[-/-] and Dnmt1[-/-] mutant embryos during mouse early organogenesis. **a** Table with the numbers of E8.5 embryos and cells of each genotype analysed in this study. KO refers to the mouse models used in this study, CRISPR indicates published data which was generated using zygotic CRISPR-Cas9 injections [23]. **b** Dimensionality reduction (UMAP) of the wildtype reference dataset used for assigning cell types in this study. Cells are coloured by cell type as in the original publication [25]. **c** RNA expression of *Dnmt1*, *Dnmt3a* and *Dnmt3b* for each cell type in the reference atlas (quantified at the pseudobulk level). **d** Mapping of the KO cells to the reference atlas using the matching nearest neighbours (MNN) algorithm [26]. Each plot shows the UMAP of the reference atlas as in **b**, but cells are coloured by whether they are a nearest neighbour to a cell in our wildtype (blue) or mutant (red) embryos. **e** Box plots display the log2 difference in cell type proportions between WT and KO E8.5 embryos. Each point represents a comparison of cell type proportions between a KO embryo and the average proportions in WT embryos. **f** Polar bar plots display the number of differentially expressed genes for each KO and cell type. In the top panel bar plots are coloured by cell type identity and in the bottom panel, they are coloured by whether genes are up or downregulated. **g** Bar plots display the number of downregulated (top) or upregulated (bottom) genes in the Dnmt1-/- mutants. Shown are only genes which are markers for embryonic versus extra-embryonic (ExE) tissues. **h** Bar plots display the number of DE genes in the *Dnmt1*[-/-] mutants for each cell type. Genes are grouped and coloured by the cell type that they mark in the reference atlas. Note that a gene might be a marker of multiple cell types, thus the *y*-axis is not directly comparable to **f**

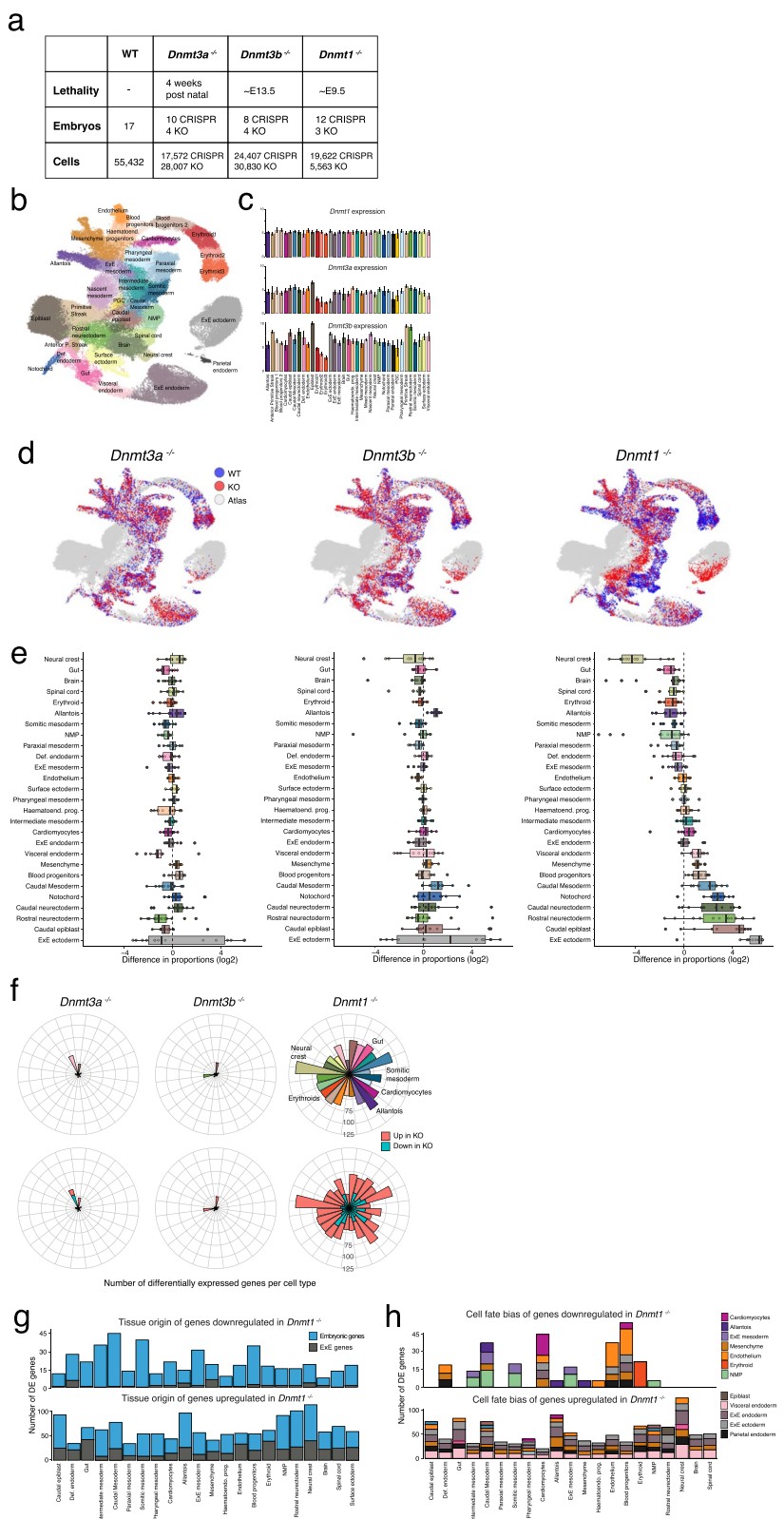

**Fig. 1** (See legend on previous page.)

Cells (ESCs) lacking DNA methylation enzymes do not differentiate efficiently and are derailed toward production of trophoblast [27–29]. We hypothesised that the underrepresentation of mature embryonic cell types could be linked to a developmental delay. To quantify this, we staged embryos by performing principal component analysis on the cell type proportions together with the reference atlas embryos that span from E6.5 to E8.5 (Additional file 1: Fig. S3, Methods). We inferred an interpretable stage assignment by measuring Euclidean distances between KO and reference embryos in the latent space. Reassuringly, we find that most embryos, including WT, $Dnmt3a^{-/-}$ and $Dnmt3b^{-/-}$ backgrounds, match the E8.25–E8.5 reference embryos with a high probability. However, $Dnmt1^{-/-}$ embryos display a minor developmental delay, and most closely resemble E8.0 reference embryos.

### DNMT1 is required for the repression of pluripotency and extra-embryonic programmes and for the up-regulation of posterior Hox genes

The profiling of large-scale single-cell transcriptomes provides sufficient statistical power to perform robust cell type-specific differential expression (DE).

First, we confirmed the upregulation of germline genes [9, 22] and dysregulation of imprints in $Dnmt1^{-/-}$ embryos [22, 30, 31]. Consistently, we find that most of these genes are misregulated across multiple cell types, albeit with some exceptions (Additional file 1: Fig. S4-5). Similarly, we confirm the upregulation of different types of repetitive elements in $Dnmt1^{-/-}$ embryos including Intracisternal A-type particles (IAPs), LINE L1 and ERVs [22, 23, 32]. Interestingly, although our results broadly agree with bulk studies, we detect cell type-specific differences in some of these elements (Additional file 1: Fig. S6).

Next, we aimed to link gene expression changes in *Dnmt* KOs to defects in cell fate commitment. We thus restricted the analysis to 2107 genes that are cell type markers in the reference data set [25]. Consistent with the cell type proportions results, we observe a small number of DE genes when comparing $Dnmt3a^{-/-}$ and $Dnmt3b^{-/-}$ to WT samples (Fig. 1e). In contrast, a larger number of DE genes is observed in the $Dnmt1^{-/-}$ across most cell types, but particularly in the Neural crest, Caudal mesoderm and Blood progenitors. In agreement with the repressive role of DNA methylation, we observe a greater number of upregulated compared to downregulated genes in $Dnmt1^{-/-}$ across most cell types (Fig. 1f).

Next, we sought to explore whether the DE genes in the $Dnmt1^{-/-}$ display enrichment towards specific cell fates. We plotted the number of DE genes for each cell type and coloured these by the cell type that each gene identifies (in the reference atlas) (Fig. 1g, h). We observe that genes downregulated in the $Dnmt1^{-/-}$ in endothelium and erythroid cells are enriched for endothelium and erythroid genes, respectively. Interestingly, genes downregulated in somitic and intermediate mesoderm cells are enriched for NMP markers and this category includes several posterior Homeobox (Hox) genes such as *Hoxc9*, *Hoxc8*, *Hoxb9* and *Hoxa9*. In the $Dnmt1^{-/-}$, these genes show significant downregulation in posterior cell types such as NMPs, somitic mesoderm, intermediate mesoderm and ExE mesoderm (Fig. 2). These Hox transcription factors display a strong transcriptomic and epigenetic signature in NMPs [33] and are essential for correct axial regionalisation. We hypothesise that the downregulation of these genes can potentially explain the

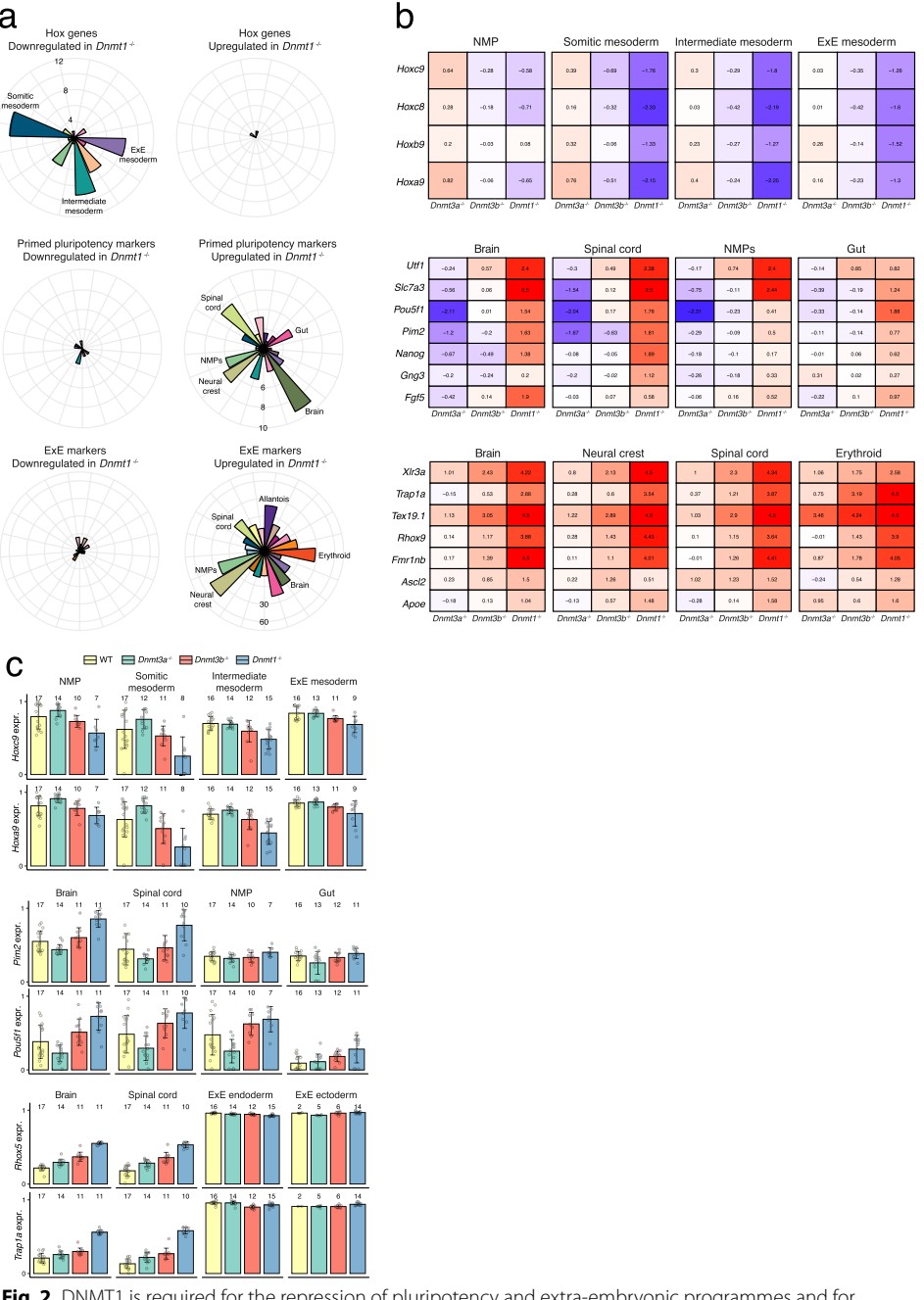

**Fig. 2** DNMT1 is required for the repression of pluripotency and extra-embryonic programmes and for the up-regulation of posterior Hox genes. **a** Polar bar plots display the number of differentially expressed genes in Dnmt1-/- cells, split by whether genes are downregulated (left) or upregulated (right). Each bar corresponds to a different cell type. Shown are all Hox genes (top) primed pluripotency markers (middle) and markers of extra-embryonic (ExE) tissues (bottom), according to the reference atlas. **b** Heatmaps display the log fold change in gene expression between mutant and wildtype. Shown are Hox genes (top), primed pluripotency markers (middle) and markers of ExE lineages (bottom). **c** Gene expression levels quantified at the pseudobulk level, where each data point corresponds to a different embryo and cell type. Shown are Hox genes (top), primed pluripotency markers (middle) and ExE tissue markers (bottom)

underrepresentation of both NMPs and its derivatives, somitic mesoderm and spinal cord cells, in *Dnmt1*^-/- embryos.

Among the genes that are upregulated in the *Dnmt1*^-/- we observe a clear enrichment for Epiblast and ExE marker genes across most cell types (Figs. 1g and 2). This includes primed pluripotency markers such as *Pou5f1*, *Utf1*, *Slc7a3*, *Fgf5* and *Pim2* for the former, and *Rhox5*, *Krt8*, *Apoe*, *Ascl2*, *Trap1a* and *Xlr3a* for the latter. Overall, these results are consistent with published bulk RNA-seq analysis of *Dnmt1*^-/- embryos [22] (Additional file 1: Fig. S7), particularly for genes with the largest log-fold differences which are differentially expressed in multiple cell types. Importantly, the bulk analysis is not able to distinguish between changes in cell type abundance (e.g. increased abundance of extra-embryonic tissue) and changes in marker gene expression (e.g. increased expression of extra-embryonic genes in embryonic tissues).

Intriguingly, both classes of genes are repressed before gastrulation in the embryo proper, but in *Dnmt1*^-/- embryos they remain expressed across multiple cell types after gastrulation. Previous studies have linked the disruption of the DNA methylation machinery with transdifferentiation events between the embryo proper and trophoblast cells. In particular, in *Dnmt1*^-/- [29] or *Dnmt3ab*^-/- (double knockout) [27] cells exiting naive pluripotency can be derailed towards a trophoblast fate and chimeric embryos generated by nuclear transfer of DNMT triple knockout cells followed by aggregation with wildtype embryos are able to form trophoblast but not embryonic lineages [28]. All together, our results support the role of DNA methylation as a repressor of past and alternative cellular identities. We hypothesise that this could be the molecular mechanism that underlies the overrepresentation of ExE tissue and the developmental delay of *Dnmt1*^-/- embryos.

### TET enzymes are required for the specification of primitive erythrocytes

We next investigated the role of active DNA demethylation by perturbation of the three TET enzymes. Due to the severity of the phenotype of embryos lacking all three TETs at E8.5 [21], we instead generated chimeric embryos from *Tet* triple knockout (TKO) ES cells [34]. In contrast to *Dnmt3a*^-/-*Dnmt3b*^-/- double knockout cells [27] and *Dnmt1*^-/- cells [29] which are rejected from chimeric embryos, *Tet*-TKO cells contribute with high efficiency at both E7.5 and E8.5 (Additional file 1: Fig. S8). We next performed scRNA-seq on these chimaeras following the study design of Pijuan-Sala et al. [25] in which *Tet*-TKO cells are marked by the fluorescent marker tdTomato thereby allowing the collection of two fractions using FACS: a fluorescent fraction that contains *Tet*-TKO cells and a non-fluorescent fraction that contains WT host cells (Fig. 3a, Additional file 1: Fig. S8). In total, we profiled 24,355 *Tet*-TKO cells and 52,084 WT cells.

Similar to the strategy employed for DNMT mutants, we assigned cell types by mapping cells to the reference atlas (Fig. 3b, c, Additional file 1: Fig. S9). As expected from chimaeras generated from ESC injection into blastocysts, we find no contribution of tdTomato+ cells on the trophoblast compartment (ExE ectoderm cells), and this is true for injected WT control and Tet-TKO cells. As an additional control, we re-analysed a published data set where WT ESCs cells marked by tdTomato were processed and sequenced in a similar fashion as our experimental design [35]. Reassuringly, negligible

differences in cell type proportions are observed when comparing (injected) tdTomato+ WT cells and (host) tdTomato− WT cells (Additional file 1: Fig. S9). This indicates that there are no major cell type biases in the contribution of injected ESCs to chimeric embryos. After the control experiments, we compared the cell type proportions between *Tet*-TKO and WT populations. We find a marked depletion of erythroid and neural crest cells in *Tet*-TKO cells at E8.5, together with an increase in mesodermal progenitor cells (mixed mesoderm, intermediate mesoderm) and ExE mesodermal tissue (mesenchyme, allantois, ExE mesoderm). The depletion of Erythroid cells in the *Tet*-TKO embryos is clearly observed when mapping cells to the haemato-endothelial trajectory reconstructed from the reference atlas [25].

Next, we staged the embryos using the same strategy as for the Dnmt KOs. As expected from the differences in cell type proportions, we infer that E8.5 *Tet*-TKO embryos display a slight delay and match E8.25 reference embryos with a higher probability (Additional file 1: Fig. S9). Nevertheless, this is not sufficient to explain the depletion of erythroid cells, which are already present in significant proportions by E8.0 in WT conditions [25].

Finally, we performed cell type-specific DE. As in our previous approach, we restricted the analysis to genes that are cell type markers in the reference data set [25]. Across most cell types, the majority of DE genes were found to be downregulated in the *Tet*-TKO (Fig. 3d), as might be expected from cells with the inability to demethylate gene regulatory elements [36]. Consistent with previous studies on *Tet*-TKO mutants, we observe diminished expression of *Lefty2* in the nascent Mesoderm (Additional file 1: Fig. S10), which results in a gain-of-function of Nodal signalling [21]. This however does not lead to major defects in early mesodermal lineages. Instead, we find that late mesodermal cell types display the highest number of DE genes, including cardiomyocytes, endothelium and erythroid cells (Fig. 3d). Of the DE genes upregulated in *Tet*-TKO, we find a number of fibroblast growth factor (FGF) genes, including *Fgf8* in nascent mesoderm cells and *Fgf3* in erythroid cells (Additional file 1: Fig. S10). FGF signalling is known to inhibit primitive blood formation in frog [37, 38] and chicken [39] embryos so its upregulation in *Tet*-TKO fits with the phenotype we observe. Notably, most of the genes that are DE in Blood progenitors and Erythroid cells have a known role in blood differentiation, such

(See figure on next page.)

**Fig. 3** scRNA-seq of Tet-TKO mutant embryos during mouse early organogenesis reveals that TET enzymes are required for the specification of primitive erythrocytes. **a** Schematic summarising the chimaera assay. Fluorescently labelled Tet-TKO ESCs are injected into wild type blastocysts, transferred into pseudopregnant hosts then collected at E7.5 or E8.5. FACS is used to isolate labelled KO cells (red) and non-labelled WT host cells (blue) which are processed and sequenced using scRNA-seq. **b** Mapping of the KO cells to the reference atlas using the matching nearest neighbours (MNN) algorithm [26]. UMAP plot of wildtype reference atlas [25] with cells coloured whether they are a nearest neighbour to a WT host (red) or Tet-TKO (blue) cell. **c** Box plots display the log2 difference in cell type proportions between WT and Tet-TKO E8.5 embryos. Each point represents a comparison of proportions between a Tet-TKO sample and the corresponding proportions in the matching WT host embryo. **d** Polar bar plots display the number of differentially expressed genes for each KO and cell type. In the right panel bar plots are coloured by cell type identity and in the left panel, they are coloured by whether genes are up or downregulated. **e** Bar plots display the number of DE genes for each cell type. Genes are grouped and coloured by the cell type that they mark in the reference atlas. Note that a gene might be a marker of multiple cell types, thus the values in the *y*-axis are not directly comparable to **d**. **f** RNA expression levels of the haemoglobin X alpha-like embryonic chain gene (Hba-x) in WT to Tet-TKO cells. Shown are different cell types grouped from the haematoendothelial trajectory

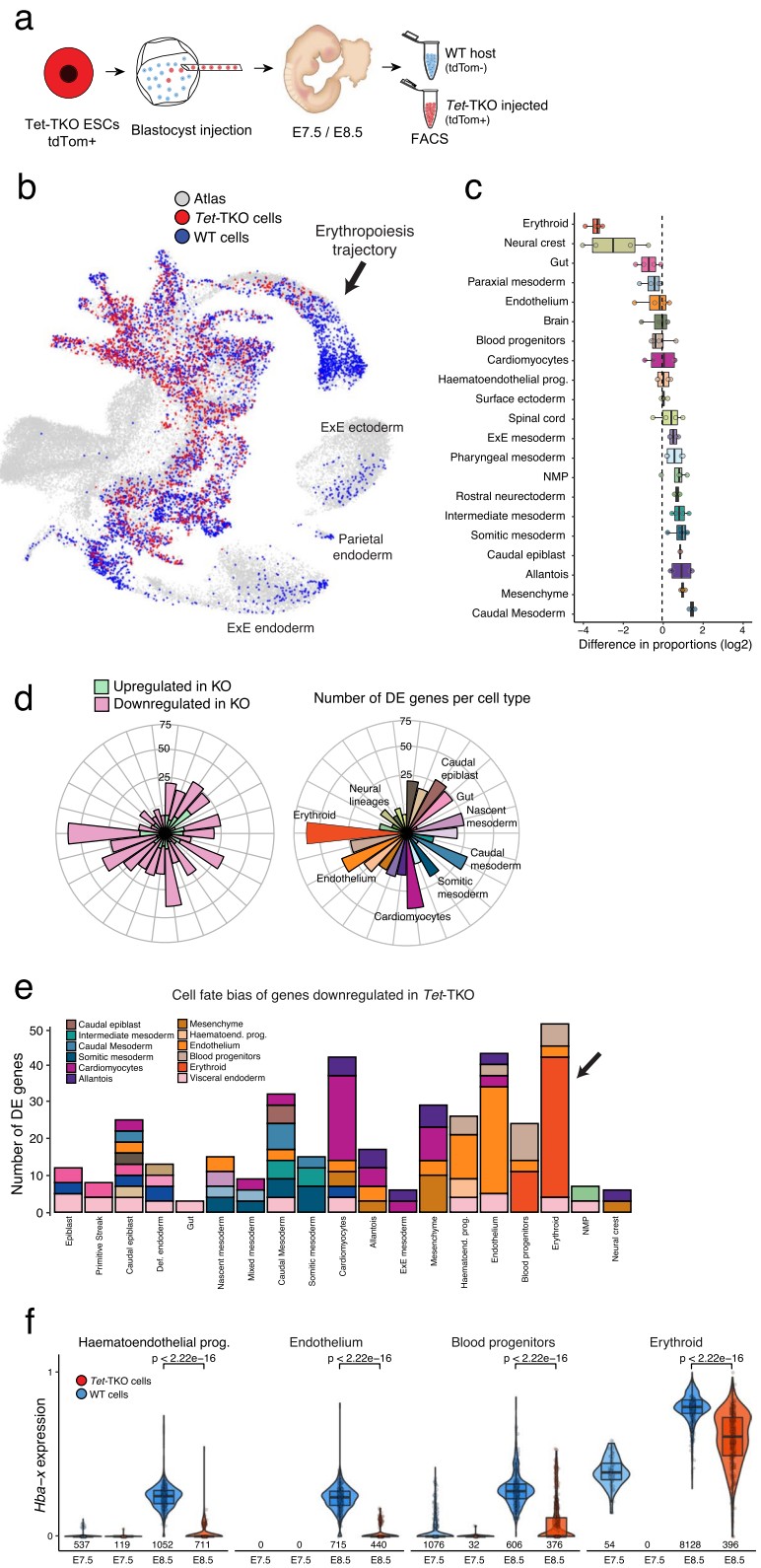

**Fig. 3** (See legend on previous page.)

as *Hba-x*, *Klf1*, *Gata1*, *Gata2*, *Hemgn* and *Alas2* (Fig. 3e, f, Additional file 1: Fig. S10). This suggests that TET enzymes are required for the up-regulation of the gene expression program that initiates blood differentiation, presumably via demethylation of these genes' regulatory regions.

### Impaired primitive erythropoiesis in *Tet*-TKO cells is linked to TET-dependent DNA demethylation of lineage-specific cis-regulatory elements

We next sought to explore how impaired demethylation might be driving the failure to form primitive blood cells in *Tet*-TKO embryos. To our knowledge, DNA methylation has never been profiled during primitive erythropoiesis. However, previous studies have reported a global loss of DNA methylation during definitive erythropoiesis [40]. The decreased expression of DNMTs along this trajectory and the requirement for DNA replication [40] suggested that this phenomenon is driven by passive DNA demethylation. However, given the phenotype we observe in *Tet*-TKO embryos, we hypothesised the involvement of the TET-dependent DNA demethylation pathway.

To explore this, we isolated specific cell populations from the haemato-endothelial trajectory in E7.5 and E8.5 WT and *Tet*-TKO backgrounds and performed single-cell multi-omics profiling of RNA expression, DNA methylation and chromatin accessibility from the same cell using scNMT-seq [24] (Fig. 4a). We sequenced 768 cells using scNMT-seq together with an additional 1056 cells using only scRNA-seq. The increased sample size of scRNA-seq data was used to aid cell type annotation. In total, 1634, 724 and 616 cells passed quality control thresholds for RNA expression, DNA methylation and chromatin accessibility, respectively (Additional file 1: Fig. S11). Cell type labels were again assigned by mapping to the reference atlas using the RNA modality (Additional file 1: Fig. S12). Reassuringly, cell types recovered matched the expectation based on the markers used (Fig. 4a, Additional file 1: Fig. S12). In spite of the vastly decreased numbers of erythroid cells in the *Tet*-TKO background, the sorting strategy allowed us to recover the entire blood trajectory in the knockout (Fig. 4a, Additional file 1: Fig. S12).

(See figure on next page.)

**Fig. 4** scNMT-seq of *Tet*-TKO cells reveals impaired DNA demethylation of erythroid enhancers during primitive erythro- poiesis. **a** Schematic summarising the scNMT-seq chimaera assay. Fluorescently labelled Tet-TKO ESCs are injected into wild type blastocysts, transferred into pseudopregnant hosts then collected at E8.5. FACS is used to isolate specific populations (CD41+, erythroid; KDR+, Haematoendothelial progenitors; CD41+ KDR+, blood progenitors and CD41−, KDR−) of both labelled KO cells (red) and non-labelled WT host cells (blue) which are processed and sequenced using scNMT-seq. **b** Scatter plot displaying expression levels of the haemoglobin alpha adult chain 1 gene (*Hba-a1*) in cells ordered along a reconstructed primitive erythropoiesis trajectory. Cells are coloured by genotype, WT (*N*=301, top) and *Tet*-TKO (*N*=221, bottom). The line displays the LOESS curve. **c** As **b** for *Dnmt* and *Tet* genes, and *Uhrf1*. To avoid cluttering the LOESS curves are shown without the corresponding data points. **d** Scatterplot displaying global CpG methylation in cells ordered along the same pseudotime trajectory as **b** and coloured by genotype. The line displays the LOESS curve. **e** DNA methylation (yellow) and chromatin accessibility (green) profiles quantified over multiple genomic contexts in WT (*N*=67,top) and *Tet*-TKO (*N*=57, bottom) erythroid cells. Each column corresponds to a different genomic context: promoters (*N*=18,329), surface ectoderm enhancers (*N*=2138), haematoendothelial progenitors enhancers (*N*=3616), and erythroid enhancers (*N*=4319). Shown is the mean +/− 1 standard deviation in running averages of 50bp windows around the centre of the genomic annotation (2kb upstream and downstream). **f** Boxplots showing the distribution of DNA methylation (top) and chromatin accessibility (bottom) in erythroid cells in WT (*N*=67, blue) and *Tet*-TKO (*N*=57, red) at different genomic annotations

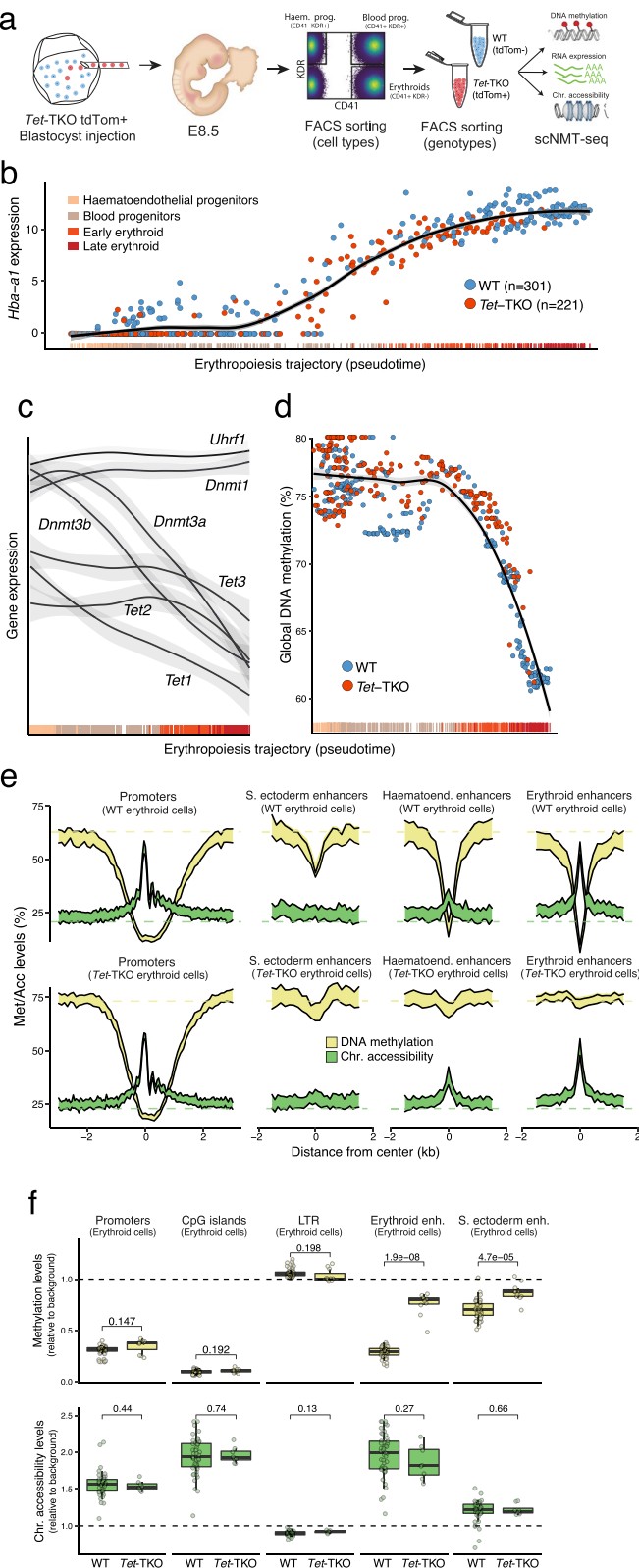

**Fig. 4** (See legend on previous page.)

Similar to definitive erythropoiesis [40], we find that the primitive erythropoiesis trajectory (Fig. 4b) is associated with a global loss of DNA methylation (Fig. 4d) and a concomitant decrease in expression of all DNA (de)methylation enzymes, except for *Dnmt1* and *Uhrf1* (Fig. 4b, c). Notably, the global loss of DNA methylation is also observed in the *Tet*-TKO cells, indicating that DNA methylation is largely lost by passive dilution during replication, possibly by downregulating protein levels of DNMT1 or UHRF1 [13] or via exclusion from the nucleus [41].

Next, we quantified DNA methylation and chromatin accessibility levels over a catalogue of distal lineage-specific regulatory elements derived from our recent multi-modal atlas of mouse early organogenesis [42], together with promoters, CpG islands and intergenic repeat elements. As expected, we find that regulatory regions associated with the blood trajectory become hypomethylated and accessible in wild type erythroid cells whereas regulatory regions associated with other lineages remain highly methylated and low in accessibility (Fig. 4e, f, Additional file 1: Fig. S13). In striking contrast, *Tet*-TKO cells remain hypermethylated at these genomic elements demonstrating that this demethylation process is TET-dependent. Interestingly, the chromatin accessibility of blood-specific regulatory regions is unchanged in the knockout cells, indicating that the two epigenetic layers are not necessarily coupled (Fig. 4e, f, Additional file 1: Fig. S13). Furthermore, TET-dependent demethylation is specific to distal regulatory regions with negligible effects at gene promoters, which retain low levels of methylation in both wild type and *Tet*-TKO cells (Fig. 4e, f, Additional file 1: Fig. S13). Notably, the same observations hold for other cell types profiled including Pharyngeal mesoderm, Surface ectoderm and ExE mesoderm (Additional file 1: Fig. S13), suggesting that TET-dependent demethylation of distal regulatory sites is a generic feature of cell fate decisions during early organogenesis. In some instances, we also observe a small reduction in the accessibility of lineage-specific sites in *Tet*-TKO cells, but these do not reach levels of regulatory regions of other lineages indicating that TET-dependent demethylation is not required for opening of enhancers. Individual representative examples of regulatory regions linked to erythropoietic genes that are differentially methylated between WT and *Tet*-TKO cells are shown in Additional file 1: Fig. S14.

All together, our results are in agreement with cell culture experiments that show an impaired differentiation potential of ESCs into embryoid bodies [43] and a failure to demethylate enhancers [36]. Additionally, work in zebrafish has also demonstrated TET-dependent de-methylation of enhancers during the pharyngula stage of development (corresponding to E9.5 in mouse) [44]. More generally, our data indicate that cell fate decisions of early organogenesis are underpinned by epigenomic changes in regulatory elements that occur in a two-step process. In a first step, chromatin is remodelled to allow accessibility to the DNA, which is followed by TET-dependent removal of DNA methylation. Following our results, we hypothesise that the first step is sufficient to initiate erythropoiesis, but the second step is required to establish erythroid identity.

## Discussion

We generated a transcriptomic atlas at single-cell resolution for Dnmt and Tet mutant mouse embryos and have made the data publicly available via an interactive platform. By mapping the gene expression profiles onto a wild-type reference we have been able to

robustly assign cell type labels and perform a comprehensive transcriptome-wide assessment of differentiation defects. The large number of embryos per genotype and the large number of cells profiled enabled us to quantify variations in cell type proportions as well as cell type-specific gene expression differences.

We find that DNA methyltransferases are dispensable for the formation of all major cell types up to E8.5. However, *Dnmt1*$^{-/-}$ embryos are developmentally delayed and fail to correctly repress primed pluripotency markers indicating that DNA methylation is required for the suppression of previous fates. We also observe an over-expression of extra-embryonic genes consistent with chimaera experiments in which *Dnmt* mutant cells transdifferentiate to the trophoblast lineage [27–29]. This fits with the lower CpG methylation levels of the extra-embryonic tissues [45], indicating that high methylation in the epiblast is used to suppress the trophoblast fate.

*Tet*-TKO embryos displayed pronounced lineage biases, in particular a disruption of primitive erythropoiesis. This is consistent with recent work that found that loss of all three Tet enzymes immediately after gastrulation display severe defects in the specification of haematopoietic stem and progenitor cells [46]. Using single-cell multi-omics technologies, we find that primitive erythrocytes are associated with global methylation loss, independent of TET enzymes, likely mirroring the demethylation that occurs later in development during definitive erythropoiesis [40]. Beyond this passive process, we now reveal coordinated demethylation of distal regulatory elements associated within the blood lineage that is TET-dependent and which provides a molecular explanation for the *Tet*-TKO phenotype. We further show that TET-dependent demethylation of distal regulatory elements is a common feature of differentiation during early organogenesis.

## Conclusions

In summary, these data provide novel insights into the role of DNA methylation during mouse development and a resource for the epigenetics and developmental biology communities.

## Methods

### Mouse models

All mice used in this study were bred and maintained in the Babraham Institute Biological Support Unit. Animal experimentation was approved by the Babraham Institute Animal Welfare and Ethical Review Body and complied with existing European Union and the UK Home Office legislation and local standards.

Mice heterozygous for mutations in *Dnmt1* [11] were crossed by natural matings and *Dnmt1*$^{-/-}$ and *Dnmt1*$^{+/+}$ embryos collected. Similarly, mice heterozygous for *Dnmt3a* [47] and *Dnmt3b* [48] were crossed to produce *Dnmt3a*$^{-/-}$ and *Dnmt3b*$^{-/-}$ with matching wildtypes.

### *Generation of H2B-tdTomato-labelled Tet-TKO ESCs*

*Tet*-TKO ESCs [34] were maintained in 2i LiF culture conditions as previously described [49]. The cell line was transfected with a CAG-driven H2B-tdTomato-IRES-Puromycin plasmid for continuous labelling with histone H2B-tdTomato using

Lipofectamine 2000 transfection reagent (Thermo Fisher Scientific, 11668019), following the manufacturer's protocol and selected with puromycin (2 μg/ml).

### Generation of Tet-TKO chimaeras

E3.5 embryos were collected from natural mating of wild-type C57BL/6J mice (Babraham Institute; Biological Support Unit (BSU)). Twelve H2B-tdTomato labelled *Tet*-TKO ESCs [34] were injected into the blastocoel and cultured for 2 h in KSOM media [50] at 37°C, 5% CO2. The chimaera blastocysts were surgically transferred into the uterus of pseudo-pregnant CD1 recipients and chimeric embryos were collected and characterised at E7.5 and E8.5.

### Single-cell isolation

Knockout mice were genotyped by PCR using tissue from the ecto-placental cone. Single embryos were dissociated into single cells using 200μl of TriplE Express for 10 min at 37°C on a shaking incubator then quenched with 1ml of ice-cold 10% FCS in PBS. Cells were filtered using a 40-μM Flowmi cell strainer, span down at 300g for 5 min then resuspended in 50μl of PBS containing 0.04% BSA. Cells were counted and viability was assessed using trypan blue staining on a Countess II instrument (Invitrogen). >95% of cells were negative for trypan blue indicating high sample quality. For chimaera experiments, embryos were pooled and dissociated as above then flow-sorted using the BD Influx High-Speed Cell Sorter (BD Biosciences) or a BD FACSAriaTM system (BD Biosciences) in a biosafety cabinet, collecting DAPI negative singlets into two 1.5-ml tubes, one for tomato positive (knockout cells) and one for tomato negative (host cells). Cells were spun down at 300g for 5 min and resuspended in 50 μl of PBS containing 0.04% BSA then counted as above.

### Single-cell RNA sequencing

scRNA-seq was performed using 10x Genomics 3′ v3 following the manufacturer's instructions and loading 16,000 cells. Sequencing was performed using an Illumina Novaseq using the
  recommended read lengths.

### scNMT-seq

Cells were stained with PE/Cyanine7 anti-mouse CD309 (KDR, Biolegend, cat 136414), CD41-BV421 and DAPI then flow-sorted into 96w plates. Only DAPI negative singlets were collected. Plates were immediately incubated with GpC methylase at 37C for 15 min to label accessible chromatin then frozen down at −80°C after adding 5μl of RLT plus buffer (Qiagen). Note that a subset of cells (128 out of 768) did not receive GpC methylase treatment in order to produce higher coverage methylation data (i.e. using scM&T-seq [51]). Plates were processed using the published protocol for scNMT-seq [52]. RNA-seq libraries were sequenced using a Nextseq 500 instrument using 75bp single-end read lengths. BS-seq libraries were sequenced using a Novaseq 6000 instrument using 150bp paired-end reads.

### CRISPR KOs

CRISPR KO data was downloaded from GSE137337 and processed together with the KO mouse lines as outlined below.

### scRNA-seq data processing

10x Genomics data pre-processing: raw files were processed with Cell Ranger 5.0.0 using default mapping arguments. Reads were mapped to the mm10 genome and counted with GRCm38.92 annotation, including tdTomato sequence for chimaera cells. Low-quality cells were filtered based on the distribution of QC metrics. For the $Dnmt^{-/-}$ and the Tet-TKO scRNA-seq data sets, cells were required to have at least 1500 UMIs, a maximum percentage of reads mapping to mitochondrial genes of 30% and a maximum percentage of reads mapping to ribosomal genes of 35%. The RNA expression of the Tet-TKO scNMT-seq cells was sequenced using Smart-seq2 [53], which yields higher coverage than 10x Genomics 3′. Thus, cells were required to have at least 4000 reads, a maximum percentage of reads mapping to mitochondrial genes of 10% and a maximum percentage of reads mapping to ribosomal genes of 20%. Finally, cells were normalised using the *scran* R package [54]. Raw counts for each cell were divided by their size factors, and the resulting normalised counts were used for further processing.

### scNMT-seq data processing

scNMT-seq data was processed as previously [3]. Briefly, HiSat2 v.2.1.0 [55] was used to align RNA-seq reads to the GRCm38 mouse genome then a count matrix generated using featureCounts [56] with the Ensembl gene annotation37 (v.87). Bismark v0.23.1 [57] was used to align DNA reads to the bisulfite converted GRCm38 mouse genome then perform methylation calling and CpG - GpC splitting. Following our previous approach [3, 24], binary methylation rates were estimated for each individual CpG or GpC site in each cell. Low-quality cells were excluded based on (1) coverage (at least 5000 CpGs for methylation data and 10,000 GpCs for accessibility data) and (2) global methylation values (at least 50% for endogenous CpG methylation and between 10 and 40% for GpC accessibility). When aggregating over genomic features (i.e. promoters, enhancers), CpG methylation and GpC accessibility rates were computed assuming a binomial model, with the number of trials being the number of observations and the number of successes being the number of methylated sites. Notably, this implies that DNA methylation and chromatin accessibility are quantified as a rate (or a percentage).

### Mapping to the reference atlas and transfer of cell type labels

Cell types were assigned by mapping the RNA expression profiles to a single-cell reference atlas from the same stages [25] by matching mutual nearest neighbours [26]. First, count matrices from both data sets were concatenated and normalised together. Highly variable genes were identified and used as input for principal components analysis. Subsequently, batch correction was applied to remove the technical variability between query and atlas cells. Then, a k-nearest neighbours (kNN) graph was computed using all cells together. For each query cell, the cell type was selected as the mode from a Dirichlet

distribution given by the cell type distribution of the top 30 nearest neighbours in the atlas (i.e. majority voting).

To visualise the mapping results, we plotted the reference UMAP from [25] and used the joint kNN graph to highlight the atlas cells that are nearest neighbours to the query cells.

### Pseudobulk

To improve the signal-to-noise ratio we derived pseudobulk replicates for each cell type and genotype. Read counts were aggregated for each group and normalised using DESeq2 [58]. Importantly, the pseudobulk representation was used to visualise average gene expression levels, but it was not used to perform statistical testing in differential expression analysis. The Integrative Genomics Viewer [59] was used to visualise pseudobulk data.

### Differential RNA expression

DE analysis was performed using the negative binomial model with quasi-likelihood test implemented in edgeR. Significant hits were called with a 1% FDR (Benjamini–Hochberg procedure) and a minimum log2 fold change of 1.

### Identification of marker genes in the reference atlas

Cell type-specific marker genes were identified based on the reference atlas. First, we performed DE analysis between each pair of cell types using the strategy outlined above. Then, for each cell type, we labelled as marker genes those that are DE in more than 75% of the comparisons.

### Embryo staging

We staged the embryos by performing principal component analysis on the cell type proportions together with the reference embryos. Then, we measured euclidean distances between KO and WT embryos in the PCA space. Finally, we obtained a probabilistic cell type stage assignment by taking the inverse of the distance and performing minmax normalisation.

### Pseudotime analysis

The pseudotime order for the erythropoiesis trajectory was inferred using diffusion maps with the *destiny* R package (v3.8.1) [60].

### Supplementary Information

Additional file 1: Figure S1. General statistics and quality control metrics for Dnmt1$^{-/-}$, Dnmt3a$^{-/-}$, Dnmt3b$^{-/-}$ and WT scRNA-seq libraries. Figure S2. Cell type assignments for Dnmt1$^{-/-}$, Dnmt3a$^{-/-}$, Dnmt3b$^{-/-}$ and WT embryos. Figure S3. Inference of embryonic stage for Dnmt1$^{-/-}$, Dnmt3a$^{-/-}$ and Dnmt3b$^{-/-}$ embryos. Figure S4. Expression changes of imprinted genes in Dnmt1$^{-/-}$, Dnmt3a$^{-/-}$ and Dnmt3b$^{-/-}$ embryos. Figure S5. Expression changes of germline genes in Dnmt1$^{-/-}$, Dnmt3a$^{-/-}$ and Dnmt3b$^{-/-}$ embryos. Figure S6. Expression changes of repetitive elements in Dnmt1$^{-/-}$, Dnmt3a$^{-/-}$ and Dnmt3b$^{-/-}$ embryos. Figure S7. Comparison of differential expression changes with a published bulk RNA-seq study. Figure S8. Overview of the Tet-TKO chimaera assay. Figure S9. Mapping, cell type assignments and embryo staging for Tet-TKO scRNA-seq samples. Figure S10. Differential gene expression analysis between WT

and Tet-TKO embryos. Figure S11. Quality control (QC) metrics for scNMT-seq Tet-TKO embryos. Figure S12. Cell type assignments of WT and Tet-TKO scNMT-seq cells. Figure S13. DNA methylation and chromatin accessibility at promoters and lineage-specific enhancers for different cell types in the Tet-TKO scNMT-seq experiment. Figure S14. Examples of individual cis-regulatory regions that are dsyregulated in Tet-TKO cells.

Additional file 2. Review history.

### Acknowledgements
We thank Paula Kokko-Gonzales, Nicole Forrester and Amelia Edwards of the Babraham Institute Sequencing Facility for assistance with 10x Genomics library preparation and Illumina Sequencing; members of the CRUK-CI Genomics Core for Illumina sequencing, members of the Babraham Flow Cytometry Core Facility for cell sorting and the Babraham Biological Support Unit for animal work; Carolina Guibentif for experimental support; Renee Beekman for discussions on the interpretation of the results; all members of the Reik lab for discussions and support.

### Peer review information

### Review history
The review history is available as Additional file 2.

### Authors' contributions
S.J.C., T.L., R.A. and W.R. conceived the project. S.J.C., T.L., D.D, and J.N. performed embryo dissections and single-cell isolation. S.J.C. performed scNMT-seq library preparation. F.K. processed and managed sequencing data. B.G. and J.C.M. provided discussion and interpretation of the data analysis. R.A. and S.J.C. performed pre-processing and quality control. R.A. performed the computational analysis and generated the figures. R.A. S.J.C., T.L. and W.R. interpreted the results and drafted the manuscript. W.R. supervised the project. All authors read and approved the final manuscript.

### Authors' information
Twitter handles: @RArgelaguet (Ricard Argelaguet); @ReikLab (Wolf Reik)

### Funding
The following sources of funding are gratefully acknowledged. This work was supported by the Wellcome Trust (awards 210754/Z/18/Z and 220379/Z/20/Z) and the BBSRC (award BBS/E/B/000C0421). T.L. was funded by the Wellcome Trust 4-Year PhD Programme in Stem Cell Biology and Medicine and the University of Cambridge, UK (203813/Z/16/A and 203813/Z/16/Z). J.N. was supported by core funding by the MRC and Wellcome Trust to the Wellcome–MRC Cambridge Stem Cell Institute. The funding sources mentioned above had no role in the study design, in the collection, analysis and interpretation of data, in the writing of the manuscript and in the decision to submit the manuscript for publication. This research was funded in whole or in part by the Wellcome Trust. R.A. was supported by the Wellcome for a Collaborative Award in Science (award 220379/Z/20/Z).

### Availability of data and materials
Code to reproduce the results in this manuscript is available via GitHub repositories for each of the three sets of analysis: DNMT scRNA-seq [61], *Tet*-TKO scRNA-seq [62] and *Tet*-TKO scNMT-seq [63]. Stable versions of all three repositories have been archived on Zenodo under a MIT Licence [64].
Raw sequencing data together with processed files are available in the Gene Expression Omnibus under accession GSE204908 [65]. Links to processed data objects as well as to an interactive R Shiny app are available in the corresponding GitHub repositories.

## Declarations

### Ethics approval and consent to participate
Animal experimentation was approved by the Babraham Institute Animal Welfare and Ethical Review Body and complied with existing European Union and the UK Home Office legislation and local standards.

### Competing interests
W.R. is a consultant and shareholder of Cambridge Epigenetix. S.J.C., R.A., D.D., F.K. and W.R. are employees of Altos Labs. The remaining authors declare no competing financial interests.

### Author details
[1]Epigenetics Programme, Babraham Institute, Cambridge CB22 3AT, UK. [2]Altos Labs Cambridge Institute of Science, Granta Park, Cambridge, UK. [3]Wellcome Trust – Medical Research Council Stem Cell Institute, University of Cambridge, Jeffrey Cheah Biomedical Centre, Puddicombe Way, Cambridge CB2 0AW, UK. [4]Bioinformatics Group, Babraham Institute, Cambridge CB22 3AT, UK. [5]Department of Haematology, Cambridge Institute for Medical Research, University of Cambridge, Cambridge, UK. [6]Cancer Research UK Cambridge Institute, University of Cambridge, Cambridge, UK. [7]European Molecular Biology Laboratory, European Bioinformatics Institute, Wellcome Genome Campus, Cambridge, UK. [8]Wellcome Sanger Institute, Wellcome Genome Campus, Cambridge, UK. [9]Department of Physiology, Development and Neuroscience, University of Cambridge, Tennis Court Road, Cambridge CB2 3EG, UK. [10]Current address: MRC Human Genetics Unit, University of Edinburgh, Crewe Road, Edinburgh EH4 2XU, UK. [11]Centre for Trophoblast Research, University of Cambridge, Cambridge, UK.

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

## 
