## [Additional file 2. Review history. · Genome Biology]

Review History

First round of review

Reviewer 1

Are you able to assess all statistics in the manuscript, including the appropriateness of statistical tests used? No, I do not feel adequately qualified to assess the statistics.

Comments to author:

DNA methylation catalyzed by DNMT enzymes and removed by TET proteins is essential for embryonic development. Inactivation of these proteins in the mouse leads to severe developmental defects, but it is not yet fully understood why these embryos die. The manuscript by Lohoff et al. addresses this question at the single cell level by performing elegant single cell transcriptomic and epigenetic analyses in a variety of Dnmt and Tet-mutant mouse embryos. Because previous studies mainly used bulk transcriptomic analysis, it is timely to apply single cell approaches towards a better understanding of the precise role of DNA methylation in mammalian embryogenesis and cell fate decisions. Thus the manuscript is of considerable interest in the field of developmental epigenetics. In particular, the last finding that TET-dependent demethylation is specifically required to generate primitive erythrocytes is interesting and could not have been discovered by bulk analyses.

Comments:

-A previous study (Grosswendt et al., Nature 2020, ref 20) reported a single cell RNA-seq analysis of Dnmt3a, Dnmt3b and Dnmt1 CRISPR-KO embryos. This work should be acknowledged in the introduction. Furthermore the authors merged their data with these previous data (Line 81-82), which is a bit intriguing. Before merging the datasets, it would be necessary to know whether both studies independently agree on the defects in cell type composition in the Dnmt mutant embryos.

-Bulk RNA-seq data exist in Dnmt1^{-/-} embryos (Dahlet et al., PMID 32561758). As an independent validation, is it possible to verify that some of the genes misregulated in the single cell analysis are also found misregulated, maybe at low levels, in bulk analysis?

-In absence of DNA methylation data, the link between Dnmt1 and the misregulated genes is unclear and could be indirect. To gain mechanistic insights, it would be necessary to correlate DNA methylation and gene expression in Dnmt1^{-/-} embryos.

-Germline genes, imprinted genes and TEs such as IAPs are the major targets of Dnmt1 in embryos, yet they seem to be excluded from the analysis in this manuscript. It would be very interesting to study whether these genes/TEs are overexpressed in all or specific cell types of the embryo to determine if DNA methylation is a universal repression mechanism.

-Figure 4a: it is unclear from this figure that specific hematopoietic cells have been isolated prior to scNMT-seq analysis. Please clarify the figure.

-Depending on the resolution of scNMT-seq, is it possible to show examples of DNA methylation, chromatin accessibility and expression of individual erythropoietic genes in WT and Tet-TKO backgrounds?

-Lines 246-250: The work by Bogdanovic et al. (PMID: 26928226) could be cited here.

Reviewer 2

Are you able to assess all statistics in the manuscript, including the appropriateness of statistical tests used? No, I do not feel adequately qualified to assess the statistics.

Comments to author:

In this study Reik and colleagues conduct single cell transcriptomic and epigenomic analyses of Dnmt and Tet mutant embryos at E8.5. First they analyze Dnmt1, 3a and 3b single knockouts and find that only Dnmt1 KO embryos are developmentally delayed and have underrepresentation of mature embryonic cell types and overrepresentation of ExE cells and genes/programs. They conclude that Dnmt1 is important for proper silencing of early developmental programs and activation of mature embryonic lineages. In a second set of experiments they analyze the developmental potential of Tet-TKO ESCs (in the context of a chimera experiment) and subject the cells to single cell expression, methylation and accessibility assays (scNMT-seq). They find underrepresentation of neural crest and erythroid lineages and overrepresentation of ExE mesoderm cells. They further find that Tet loss leads to hypermethylation of developmental enhancers without much effects on accessibility. They conclude that Dnmt1 and Tets work to silence early program and activate mature embryonic programs. They nicely reference prior work that practically validates their findings (such as limitation of Dnmt1 KO cells to contribute to development and requirement of Tets in hematopoiesis as reported by genetic studies).

The study is well designed and data is clearly presented. The findings are very interesting, timely and suitable for publication in Genome Biology. In particular the scNMT-seq of Tet TKO cells is very new and provides nice insights into molecular requirements of these enzymes in mid-gestation development. There is some lack of explanation of some figures in the results section and some points require more discussion (which are noted below). With these minor changes/clarifications the study is suitable for publication and will be a very nice contribution to the field.

Minor points:

1. They indicate that the rationale for pooling their data with published Crispr KO data is to increase power. I wonder what the results would be if the two data sets were analyzed separately. Specially the published crispr data has a lot of cells in their data sets. It would be good to comment on this point in the manuscript (no new analyses needed). Did the other study that used CRISPR KO do similar analysis for E8.5? If so what were their findings and can you elaborate

on those or discuss that study in the discussion in a little more detail.

2. In Figure 1, Why neural crest genes upregulated, when neural crest cell types are underrepresented?

3. In Fig S6B Gata1 down regulation should be emphasized in text (since it is relevant to erythropoiesis). Runx1 is up in Tet TKO in the same figure which is not quite clear to me given its involvement regulation of hematopoiesis (it might be useful to plot Runx1 expression like it is done for Gata1?) and comment on it in discussion.

4. In Figure S8 there is some reverse correlation between methylation and accessibility which is not quite explained in the text. Mostly the lack of correlation between methylation and accessibility is emphasized in line 240 related to erythroid lineage. It is important to explain figure S8C in detail in text.

5. In line 144, the experiment of chimeric Dnmt1 embryos: this experiment is dnmt KO NT followed by aggregation with WT embryos to make chimeras. This should be stated correctly else the sentence is confusing as to how NT can give chimeras.

6. Line 153, the reason for why germline Tet TKO were not used should be that these mice fail to develop to E8.5 for analysis. The current reason given "to avoid disruption of zygotic demethylation" is somewhat weak (What if disruption of zygotic demethylation has implication on development later).

7. Line 156, the Dnmt TKO ESC leading to developmental defects (is this data from their previous paper? That should be cited or other reference cited. Maybe a reference is missing (as I don't find any data in this study related to Dnmt TKO chimera).

8. The Tet TKO not contributing to extra-embryonic tissues to be discussed in the context of other literature where some Tet KOs are shown to contribute to extra embryonic cells.

9. In line 217, for the scNMTseq how many cells were used? (only cells that passed the test are shown currently).

10. In Figure S5A two WTs contribute to ExE ectoderm (is there some variability between the embryos?)

11. Throughout manuscript use "active DNA demethylation" instead of "active demethylation".

12. Use "Dnmt and Tet mutant mouse embryos" instead of "a variety of DNA methylation mouse embryos"

13. In Line 39, and few other places, it is presumed that Tet mediated DNA demethylation leads to active demethylation only. Tet-mediated hydroxylation can drive passive demethylation by Uhrf1 evasion. This should be corrected in the manuscript.

14. Please include reference for Tet TKO ESCs in the methods section (it is noted in the results but not in methods)

Reviewer 3

Are you able to assess all statistics in the manuscript, including the appropriateness of statistical tests used? Yes, and I have assessed the statistics in my report.

Comments to author:

Summary

In this manuscript, the authors present single-cell RNA-seq data from *Dnmt1*^{-/-}, *Dnmt3a*^{-/-}, *Dnmt3b*^{-/-}, and Tet-TKO in mouse E8.5 embryos, systematically explored the transcriptomic effect of DNA methylation writer and eraser perturbations in mouse early organogenesis. *Dnmt3a*^{-/-} and *Dnmt3b*^{-/-} show minor transcriptomic impact on E8.5, while *Dnmt1*^{-/-} shows developmental delay and over-expression of pluripotency and extra-embryonic genes. Tet-TKO shows biased lineage with the largest reduction in Erythroid and Neural crest. To provide additional evidence on Tet-TKO impacting the erythropoiesis, the authors also used their multi-omic technology, scNMT-seq, to further profile haemato-endothelial trajectory cells from Tet-TKO embryos. The authors found that TET-dependent demethylation in lineage enhancers is independent of chromatin accessibility changes, suggesting that active demethylation is required in erythropoiesis. Overall, this manuscript comprehensively discusses the DNA methylation regulator functions during early organogenesis at the cell-type-specific level. The analysis is well organized and easy to follow, and the findings are intriguing. I have several comments for the authors to improve their manuscript.

Major Comments:

The authors should provide more details on the scNMT-seq dataset since the title emphasizes that this manuscript is "multi-omic" profiling:

- 1) what are the general statistics and quality control metrics (coverage, number of genes detected, overall GpC, CpG level, bisulfite non-conversion rate, etc) for the scNMT-seq cells? A supplementary figure (like Fig. S1 for scRNA-seq) is missing.
- 2) Does the author have different QC criteria for each modality of the scNMT-seq? The method does not explain how the cell numbers on Line 216 are determined.
- 3) When mapping the scNMT-seq cell to the RNA reference atlas from Ref 21, does the author only use transcriptome information? Are there cells passing Met or NOME QC but failed RNA QC? Will these cells be included?
- 4) Fig 4e, f seems to rely on an unpublished scATAC-seq study (line 234) to determine distal lineage-specific regulatory elements. Is it possible to de novo identify potential regulatory

elements with the current scNMT-seq data in this study? Such as performing differentially methylated region via CpG methylation or differentially accessible region via GpC methylation? It is hard to evaluate an unpublished dataset, yet the regions from that study are critical to the main finding of this manuscript.

5) For haematoendothelial enhancer regions shown in Fig. 4e, are these enhancer regions intergenic or intragenic? Do they locate near the promoter or gene body of DEGs in Figure 3? Can the authors further associate some of these enhancers with DEGs in Figure 3?

Minor Comments

The sorting strategy looks the same between Fig. 3a and Fig. 4a schematics, but as mentioned in the method (line 378), scNMT sorting is labeled with additional antibodies to enrich specific cell populations. Fig. 4a should reflect that difference.

Line 174, there is no Figure S5f.

How is the PAGA graph generated in Fig. 1b and Fig. S8a? Not mentioned in the method section.

How is the pseudo-time analysis done in Fig. 4? Not mentioned in the method section.

Line 228, Gene Uhrf1 is mentioned in the text but does not occur in Fig. 4c.

Line 417, In the "Cell type annotation" part, how does the query cell be mapped to the reference UMAP after assigned cell types (all the reference cell UMAP coordinates were from Ref 21)?

I have a concern about data access. Currently, all the links in the data availability section are not available to visit. In addition, a GEO accession # (or another public database accession) is also not provided.

Response to Reviewers

All rebuttal figures are available in high-resolution at <https://www.dropbox.com/sh/wscf9ueovn28ksc/AACvU4b-4pbRN2IT4Bm2t0Aba?dl=0>

Reviewer #1:

DNA methylation catalyzed by DNMT enzymes and removed by TET proteins is essential for embryonic development. Inactivation of these proteins in the mouse leads to severe developmental defects, but it is not yet fully understood why these embryos die.

The manuscript by Lohoff et al. addresses this question at the single cell level by performing elegant single cell transcriptomic and epigenetic analyses in a variety of Dnmt and Tet-mutant mouse embryos. Because previous studies mainly used bulk transcriptomic analysis, it is timely to apply single cell approaches towards a better understanding of the precise role of DNA methylation in mammalian embryogenesis and cell fate decisions. Thus the manuscript is of considerable interest in the field of developmental epigenetics. In particular, the last finding that TET-dependent demethylation is specifically required to generate primitive erythrocytes is interesting and could not have been discovered by bulk analyses.

Comments:

-A previous study (Grosswendt et al., Nature 2020, ref 20) reported a single cell RNA-seq analysis of Dnmt3a, Dnmt3b and Dnmt1 CRISPR-KO embryos. This work should be acknowledged in the introduction. Furthermore the authors merged their data with these previous data (Line 81-82), which is a bit intriguing. Before merging the datasets, it would be necessary to know whether both studies independently agree on the defects in cell type composition in the Dnmt mutant embryos.

We thank the reviewer for highlighting this omission from our introduction which we have corrected.

We merged both studies to improve statistical power and mitigate potential technical variability driven by the nature of the genetic perturbation (i.e. CRISPR/Cas9 electroporation vs mouse knockout). We agree however that a comparison is necessary. In the following plot we show the cell type proportions split by data set: CRISPR (Grosswendt et al. 2020) vs KO (This study):

Rebuttal Figure 1. Boxplots displaying the difference in cell type proportions between knockout and wildtype from the two different datasets: CRISPR (Grosswendt et al., 2020) and KO (this study). The number of cells and embryos for each data set is shown in Figure 1a.

To quantify the consistency between the two data sets, we plotted the relationship between the differential cell type proportions between the two studies:

Rebuttal Figure 2. Scatterplots showing the difference in cell type proportions between wildtype and knockout comparing our dataset (x-axis) to the published dataset (y-axis)(Grosswendt et al., 2020). Each dot is a celltype.

As expected, for Dnmt3a^{-/-} and Dnmt3b^{-/-} we observe only very small differences in cell type proportions in both studies, which are likely to be due to technical sampling when performing the embryo dissociations. For Dnmt1^{-/-}, large differences in cell type proportions are consistent between the two studies, but with some exceptions. We observe that NMPs and Brain are strongly underrepresented in the KO data set, but only slightly underrepresented in the CRISPR data set. We hypothesise that this might be attributed to the mosaicism (a mixture of mutant and WT cells in the same embryo) that can arise in CRISPR embryos, which would lead to an apparently milder phenotype.

As per the reviewer's request, we added Supplementary Figure 3 where the cell type proportions are split per data set. We also implemented this functionality in our interactive R Shiny app (https://www.bioinformatics.babraham.ac.uk/shiny/dnmt_ko_embryo_scrna/).

-Bulk RNA-seq data exist in Dnmt1^{-/-} embryos (Dahlet et al., PMID 32561758). As an independent validation, is it possible to verify that some of the genes misregulated in the single cell analysis are also found misregulated, maybe at low levels, in bulk analysis?

We thank the reviewer for this suggestion. First we would like to highlight that the main advantages of performing differential gene expression (DE) analysis at the single-cell level are (1) the ability to detect celltype-specific changes and (2) by doing so avoid confounding the DE analysis by changes

in celltype composition. Yet, the reviewer is correct that some of the hits should be consistent in both studies, particularly when the gene expression dysregulation is observed across multiple cell types (such as in germline, pluripotency and extraembryonic genes, as shown in Figure 2).

To make a genome-wide comparison of DE results with the published bulk dataset (Dahlet et al., 2020), we first compared the log-fold change values between the two analyses. Indeed we find good agreement between the single-cell analysis and the published bulk analysis, particularly for genes with the largest logFC differences (**Rebuttal Figure 3a**). We next inspected each of the significant DE genes we highlighted in Figure 2 (Hox genes, pluripotency markers and extra-embryonic markers) using the published bulk data (Dahlet et al., 2020). A number of these genes are DE in the bulk analysis (8 out of 19). Interestingly, even when not a significant hit, small differences in expression are observed in the published bulk data that agree with our analysis (for example Hoxa9, Hoxc9, Pou5f1, Apoe, etc.).

It is worth noting that in our single-cell analysis of Dnmt1^{-/-} embryos, we were able to detect both the over-expression of extra-embryonic genes in embryonic cell types (bottom panel, Figure 2) as well as detecting an over-abundance of extra-embryonic ectoderm tissues (Figure 1e right). Similarly we were also able to show that over-expression of primed pluripotency markers (middle panel Figure 2) was independent of the increase in abundance of the Epiblast celltype (Figure 1e right). In the bulk RNA-seq analysis this appears only as over-expression of these genes.

Rebuttal Figure 3.

- (a) Scatterplots showing \log_2 fold-change in gene expression between WT and Dnmt1^{-/-} embryos from published bulk RNA-seq data (Dahlet et al., 2020) (x-axis) and this study (y-axis). Each dot is a gene and genes are labelled which are significantly differentially expressed and are either Hox genes or known markers for pluripotency or extra-embryonic celltypes (i.e. the same genes shown in Figure 2). Note that our analysis was performed separately for each celltype whereas the bulk analysis was only performed once, hence the x-axis values will be the same for each sub-plot.
- (b) Barplots displaying gene expression values from published bulk data (Dahlet et al., 2020) in WT and Dnmt1^{-/-} embryos. Shown are genes featured in Figure 2 which are differentially expressed in at least one cell type in our analysis.

We now include these comparisons in a Supplementary Figure S7 and have expanded the text to highlight these results.

-In absence of DNA methylation data, the link between Dnmt1 and the misregulated genes is unclear and could be indirect. To gain mechanistic insights, it would be necessary to correlate DNA methylation and gene expression in Dnmt1^{-/-} embryos.

We agree that our analysis does not demonstrate a causative link between DNA methylation and miss-regulated gene expression. To better explore the interplay between cell type specification, RNA expression and DNA methylation we would need to simultaneously profile RNA expression and DNA methylation from single cells in whole E8.5 embryos. This would enable the correlation between RNA expression and DNA methylation across cell types (as we did in our previous study (see Figure 1g: <https://www.nature.com/articles/s41586-019-1825-8>). However, currently available technologies (i.e. scM&T-seq and others) have limited throughput, thus making the study of a whole E8.5 embryo impractical.

Another challenge of this analysis is that *Dnmt1*^{-/-} embryos are globally demethylated, and hence one cannot distinguish causal relationships between DNA methylation at individual regulatory elements and RNA expression. To illustrate this, shown below are examples of two DE genes between WT and *Dnmt1*^{-/-} (*Hoxc9* and *Pou5f1*, see Figure 2), alongside the (bulk) DNA methylation patterns (taken from Dahlet et al 2020) near its genomic loci.

Rebuttal Figure 4. Genome browser plots displaying DNA methylation values taken from published bulk BS-seq (Dahlet et al., 2020) at two loci which contain a differentially expressed gene in our analysis (*Hoxc* cluster on the top and *Pou5f1* on the bottom).

In conclusion, finding mechanistic links between DNA methylation at regulatory regions and gene expression would require modulating DNA methylation levels of specific regulatory regions using for example CRISPR/Cas9 technologies. We hope the reviewer agrees that this is beyond the scope of this manuscript.

-Germline genes, imprinted genes and TEs such as IAPs are the major targets of Dnmt1 in embryos, yet they seem to be excluded from the analysis in this manuscript. It would be very interesting to study whether these genes/TEs are overexpressed in all or specific cell types of the embryo to determine if DNA methylation is a universal repression mechanism.

We thank the reviewer for pointing this out. Indeed, we missed this analysis in our initial submission. We now include these results in the manuscript and in **Supplementary Figures 4-6**. Please see below for more detail.

Imprinted genes

First, we explored the gene expression patterns of imprinted genes. Consistent with the results of Dahlet et al. 2020 and others we find a number of imprinted genes to be dysregulated in the *Dnmt1*^{-/-}, with negligible dysregulation in the *Dnmt3a*^{-/-} and *Dnmt3b*^{-/-} embryos. Interestingly, we find a few imprinted genes to be downregulated in the *Dnmt1*^{-/-} (*Cdkn1c*, *Grb10*, *Igf2r*), but most genes displayed upregulation, as expected (*Rhox5*, *Peg3*, *Peg10*, *Xlr3b*, *H19*, *Impact*). Both results are consistent with bulk RNA expression analysis from (Dahlet et al., 2020), and are explained by direct repression of imprinted genes by DNA methylation, or their repression by linked unmethylated antisense RNA genes or silencer elements, respectively. The directional effects and magnitudes of differentially expressed imprinted genes is similar across the vast majority of cell types (except for extraembryonic tissues such as ExE endoderm and Visceral endoderm), which explains why the results are similar with bulk RNA expression analysis.

Rebuttal Figure 5 (now Figure S4). Differential expression analysis of imprinted genes in *Dnmt1^{-/-}*, *Dnmt3a^{-/-}* and *Dnmt3b^{-/-}* embryos.

(a) Heatmaps display the log fold change in gene expression between *Dnmt* mutants and WT.

(b) Gene expression levels in pseudobulked samples for each genotype. Each data point corresponds to a different embryo and cell type.

Germline genes

Second, we explored the gene expression patterns of germline genes. Again, consistent with the results of Dahlet et al. 2020 and others we find a number of germline genes to be dysregulated in the *Dnmt1^{-/-}*, with negligible defects in the *Dnmt3a^{-/-}* and *Dnmt3b^{-/-}* embryos. This includes *Asz1*, *Dazl*, *Fkbp6*, *Tex19.1*, *Tuba3b*, *Sohlh2*. In contrast to the imprinted genes, in the case of germline genes we find these genes to be exclusively upregulated. Some of these genes display a similar magnitude across most cell types (*Dazl*, *Fkbp6*, *Tex19.1*), but interestingly some genes display cell type specific effects (*Asz1*, *Tuba3b*, *Sohlh2*). Unlike imprinted genes, we do not find clear differences between embryonic and extra-embryonic genes.

Rebuttal Figure 6 (now Figure S5). Differential expression analysis of germline genes in *Dnmt1^{-/-}*, *Dnmt3a^{-/-}* and *Dnmt3b^{-/-}* embryos.

(a) Heatmaps display the log fold change in gene expression between *Dnmt* mutants and WT.

(b) Gene expression levels in pseudobulked samples for each genotype. Each data point corresponds to a different embryo and cell type.

Transposable elements (TEs) and other repeat sequences

We previously excluded all repetitive sequences from our analysis due to the challenges of performing alignments. Coupled with the generally low expression levels of these transcripts, this makes it challenging to robustly measure the expression of TEs in single cells. To overcome these limitations, we have now quantified RNA expression of repetitive sequences after (1) aggregating the reads across all repeats from the same class and (2) aggregating the reads across all cells from a single embryo that are annotated to the same cell type. Thus, in contrast to previous studies that employed bulk RNA-seq, this approach enables us to interrogate celltype-specific changes in TEs expression.

As the reviewer points out, previous studies have shown that some types of TEs (in particular IAPs) are direct targets of DNMT1, and they become expressed upon inactivation of DNMT1. Consistent with these studies, we find upregulation of several classes of TEs as well as other repetitive transcripts in the *Dnmt1^{-/-}* (see figure below). Interestingly, IAPs and minor satellites are upregulated in all *Dnmt1^{-/-}* cell types, whereas, for example LINE L1 elements are over-expressed only in certain celltypes, including extra-embryonic endoderm and rostral neuroectoderm.

-Figure 4a: it is unclear from this figure that specific hematopoietic cells have been isolated prior to scNMT-seq analysis. Please clarify the figure.

We have updated **Figure 4a** as suggested.

Rebuttal Figure 8 (and Figure 4a in the revised MS). Schematic summarising the scNMT-seq chimera assay. Fluorescently labelled Tet-TKO ESCs are injected into wildtype blastocysts, transferred into pseudopregnant hosts then collected at E8.5. FACS is used to isolate specific populations (CD41+, erythroid; KDR+, Haematoendothelial progenitors; CD41+ KDR+, blood progenitors and CD41-, KDR-) of both labelled KO cells (red) and non-labelled WT host cells (blue) which are processed and sequenced using scNMT-seq.

-Depending on the resolution of scNMT-seq, is it possible to show examples of DNA methylation, chromatin accessibility and expression of individual erythropoietic genes in WT and Tet-TKO backgrounds?

As the reviewer suggests, the resolution of the data limits our ability to inspect individual loci at the single-cell level. This analysis starts to become possible after aggregating measurements across hundreds (ideally thousands) of cells. See below for some genome browser snapshots of erythropoietic genes that are differentially expressed between WT and Tet-TKO. We include single-cell ATAC-seq data that was used to generate the lineage-specific feature sets (i.e. the blood enhancers). These data include 10's of thousands of cells from wildtype embryos (Argelaguet et al., 2022) and helps illustrate how higher coverage can improve signal. Note that the methylation signal from a few hundred cells is more difficult to interpret and therefore the the most robust strategy to quantify DNA methylation (and chromatin accessibility from our scNMT-seq dataset) is to pool across multiple related features such as celltype-specific enhancers - as in the analysis presented in Figure 4e-f and SF8.

As per the reviewer's request, we included these examples in **Supplementary Figure S14**.

Rebuttal Figure 9 (now Figure S14). Left: Genome browser plots of loci containing differentially expressed genes between wildtype and *Tet*-TKO. Right: box and violin plots displaying gene expression values (log₂ normalised) of the same genes as in left.

-Lines 246-250: The work by Bogdanovic et al. (PMID: 26928226) could be cited here.

This work has now been cited:

“Additionally, work in zebrafish has also demonstrated TET-dependent de-methylation of enhancers during the pharyngula stage of development (corresponding to E9.5 in mouse)(Bogdanović et al. 2016)”

Reviewer #2:

In this study Reik and colleagues conduct single cell transcriptomic and epigenomic analyses of Dnmt and Tet mutant embryos at E8.5. First they analyze Dnmt1, 3a and 3b single knockouts and find that only Dnmt1 KO embryos are developmentally delayed and have underrepresentation of mature embryonic cell types and overrepresentation of ExE cells and genes/programs. They conclude that Dnmt1 is important for proper silencing of early developmental programs and activation of mature embryonic lineages. In a second set of experiments they analyze the developmental potential of Tet-TKO ESCs (in the context of a chimera experiment) and subject the cells to single cell expression, methylation and accessibility assays (scNMT-seq). They find underrepresentation of neural crest and erythroid lineages and overrepresentation of ExE mesoderm cells. They further find that Tet loss leads to hypermethylation of developmental enhancers without much effects on accessibility. They conclude that Dnmt1 and Tets work to silence early program and activate mature embryonic programs. They nicely reference prior work that practically validates their findings (such as limitation of Dnmt1 KO cells to contribute to development and requirement of Tets in hematopoiesis as reported by genetic studies).

The study is well designed and data is clearly presented. The findings are very interesting, timely and suitable for publication in Genome Biology. In particular the scNMT-seq of Tet TKO cells is very new and provides nice insights into molecular requirements of these enzymes in mid-gestation development. There is some lack of explanation of some figures in the results section and some points require more discussion (which are noted below). With these minor changes/clarifications the study is suitable for publication and will be a very nice contribution to the field.

Minor points:

1. They indicate that the rationale for pooling their data with published Crispr KO data is to increase power. I wonder what the results would be if the two data sets were analyzed separately. Specially the published crispr data has a lot of cells in their data sets. It would be good to comment on this point in the manuscript (no new analyses needed). Did the other study that used CRISPR KO do similar analysis for E8.5? If so what were their findings and can you elaborate on those or discuss that study in the discussion in a little more detail.

We thank the reviewer for pointing this out, a question also raised by Reviewer 1. This is indeed an important analysis that we omitted from our first submission. We have added a Supplementary Figure 3 where the cell type proportions are split per data set, and implemented this functionality in our [interactive R Shiny app](https://www.bioinformatics.babraham.ac.uk/shiny/dnmt_ko_embryo_scrna/) (https://www.bioinformatics.babraham.ac.uk/shiny/dnmt_ko_embryo_scrna/). For more details, please see our response to the first question of Reviewer 1.

Regarding the findings published by Grosswendt et al 2020, the majority of this was focussed on polycomb mutants. The only analysis of Dnmt mutants focussed on confirming known results (activation of transposable elements, loss of imprints), hence we felt that a reanalysis of their data is merited.

2. In Figure 1, Why neural crest genes upregulated, when neural crest cell types are underrepresented?

We apologise for the confusion but this is actually not the case. What is shown in Figure 1e is the total number of differentially expressed (DE) genes for each celltype, not the number of DE *marker* genes. In the case of the neural crest, we find that *Dnmt1*^{-/-} neural crest cells have many (upregulated) DE genes compared to the WT neural crest. As we show in Figure 1f-g many of these genes are mainly markers of epiblast (pluripotency markers) as well as extraembryonic cell types (Visceral endoderm, ExE ectoderm, ExE endoderm, Parietal endoderm). Some examples are shown below:

Rebuttal Figure 10. Barplots displaying gene expression values (log2 normalised) in WT vs *Dnmt1*^{-/-} embryos for extraembryonic cell type markers. Each dot shows the gene expression values per embryo and cell type (after pseudobulk, see Methods).

Neural crest markers, such as *Foxd3*, *Tfap2a* or *Sox9*, do not change in expression between Neural crest WT and *Dnmt1*^{-/-} cells:

Rebuttal Figure 11. Barplots displaying gene expression values (log2 normalised) in WT vs *Dnmt1*^{-/-} embryos for genes found to be differentially expressed in Neural Crest cells. Each dot shows the gene expression values per embryo and cell type (after pseudobulk, see Methods).

To clarify this for the reader, we have added text to the figure and updated the figure legend.

3. In Fig S6B *Gata1* down regulation should be emphasized in text (since it is relevant to erythropoiesis). *Runx1* is up in Tet TKO in the same figure which is not quite clear to me given its involvement regulation of hematopoiesis (it might be useful to plot *Runx1* expression like it is done for *Gata1*?) and comment on it in discussion.

This is an interesting point. Indeed, we observe higher expression of *Gata1* and *Klf1* in WT erythroids, but higher expression of *Runx1* in Tet-TKO erythroids:

Rebuttal Figure 12. Box and violin plots displaying gene expression values (log2 normalised) in wildtype (red) and *Tet*-TKO (blue) cells for known blood marker genes, *Runx1* (top), *Gata1* (middle) and *Klf1* (bottom). Each column displays a different celltype from the primitive erythropoiesis trajectory. Each dot corresponds to a cell.

To address this question, we first explored the RNA expression dynamics of blood TFs in the reference atlas (Pijuan-Sala et al., 2019). *Runx1* becomes expressed at the transition from haematoendothelial progenitors (also called hemogenic endothelium) to blood progenitors. Upon formation of erythroids, *Runx1* expression decreases, in contrast to other canonical blood TFs such as *Gata1* or *Klf1*, which gain expression upon commitment to erythroid fate. Hence, *Runx1* marks blood progenitors, whereas *Gata1* and *Klf1* mark erythroids.

Rebuttal Figure 13. Pseudotime trajectory of primitive erythropoiesis taken from (Pijuan-Sala et al., 2019), with dots coloured by celltype (left) and expression of *Runx1*, *Gata1* or *Klf1*. The right plot displays expression values (log2 normalised) of the same three genes as a function of each cell's position on this same pseudo time axis.

The results presented in this manuscript suggest that TET enzymes are important for the transition from blood progenitors to erythroid cells. Thus *Tet*-TKO embryos yield few (and probably immature) Erythroids that still retain *Runx1* expression. To confirm this hypothesis, we mapped WT and *Tet*-TKO erythroid cells to the haematoendothelial trajectory from the reference atlas (see Figure 3 from (Pijuan-Sala et al., 2019)). As expected, we observe that the few erythroids identified in *Tet*-TKO cells map to the immature Erythroids that still have not fully repressed *Runx1* expression:

Rebuttal Figure 14. Celltype mapping of WT and *Tet*-TKO cells onto the pseudotime trajectory of primitive erythropoiesis taken from (Pijuan-Sala et al., 2019). Dots are coloured by celltype (left) or whether they are a nearest neighbour to cells in our wildtype (middle) or Tet TKO (right) dataset. The arrow in the right plot highlights the presence of TKO cells in the Erythroid 1 (less mature) and Erythroid 2 but not Erythroid 3 (mature) sub-populations.

4. In Figure S8 there is some reverse correlation between methylation and accessibility which is not quite explained in the text. Mostly the lack of correlation between methylation and accessibility is emphasized in line 240 related to erythroid lineage. It is important to explain figure S8C in detail in text.

The reviewer is correct that for the cell types included in the supplementary figure there is a slight reduction in chromatin accessibility in *Tet*-TKO cells. We have amended the text to clarify this. Of note, the reduction is minor and does not reduce accessibility to levels seen in the regulatory regions of other lineages - i.e. Tet is not required to open lineage specific enhancers.

We have updated the text to clarify this:

“Notably, the same observations hold for other cell types profiled including Pharyngeal mesoderm, Surface ectoderm and ExE mesoderm [Figure S14], suggesting that TET-dependent demethylation of distal regulatory sites is a generic feature of cell fate decisions during early organogenesis. **In some instances, we also observe a small reduction in the accessibility of lineage-specific sites in knockout cells, but these do not reach levels of regulatory regions of other lineages indicating that TET-dependent demethylation is not required for opening of enhancers.**”

5. In line 144, the experiment of chimeric Dnmt1 embryos: this experiment is dnmt KO NT followed by aggregation with WT embryos to make chimeras. This should be stated correctly else the sentence is confusing as to how NT can give chimeras.

We have updated the text to clarify this:

“In particular, in *Dnmt1*^{-/-} (Ng et al., 2008) or *Dnmt3a*^{-/-} / *Dnmt3b*^{-/-} double knockout (Kinoshita et al., 2021) cells exiting naive pluripotency can be derailed towards a trophoblast fate and chimeric embryos generated by nuclear transfer of *Dnmt1*^{-/-} / *Dnmt3a*^{-/-} / *Dnmt3b*^{-/-} triple knockout cells followed by aggregation with wildtype embryos are able to form trophoblast but not embryonic lineages (Sakaue et al., 2010)”

6. Line 153, the reason for why germline Tet TKO were not used should be that these mice fail to develop to E8.5 for analysis. The current reason given "to avoid disruption of zygotic demethylation" is somewhat weak (What if disruption of zygotic demethylation has implication on development later).

Good point. We have updated the text to correct this:

“Due to the severity of the phenotype of embryos lacking all three TETs at E8.5 we instead generated chimeric embryos from Tet triple knockout (TKO) ES cells.

7. Line 156, the Dnmt TKO ESC leading to developmental defects (is this data from their previous paper? That should be cited or other reference cited. Maybe a reference is missing (as I don't find any data in this study related to Dnmt TKO chimera).

We thank the reviewer for highlighting this - the text was not clear and citations were lacking. In fact, we were referring to data which show that either *Dnmt3a*^{-/-}/*Dnmt3b*^{-/-} or *Dnmt1*^{-/-} cells do not contribute to chimeras (and not Dnmt triple knockout). The text has been updated as follows:

“In contrast to *Dnmt3a*^{-/-} *Dnmt3b*^{-/-} double knockout cells (Kinoshita et al., 2021) and *Dnmt1*^{-/-} cells (Ng et al., 2008) which are rejected from chimeric embryos, *Tet*-TKO cells contribute with high efficiency at both E7.5 and E8.5 [Figure S9].”

8. The Tet TKO not contributing to extra-embryonic tissues to be discussed in the context of other literature where some Tet KOs are shown to contribute to extra embryonic cells.

We apologise that this was not properly explained in the text. The lack of contribution of injected ESCs to the extra-embryonic compartment is expected, as ESCs are derived from embryonic tissue. This is actually a control we use to validate that the chimaera experiment worked, and is independent of the *Tet* genotype. The same has been shown for other chimeras (Pijuan-Sala et al., 2019)

To illustrate this, we show below the cell type proportions at E7.5 of tdTomato⁺ and tdTomato⁻ cells when injecting *Tet*-TKO ESCs and WT ESCs. The reviewer can appreciate that in both cases tdTomato⁺ injected cells do not contribute to extraembryonic cells (ExE ectoderm, ExE endoderm, Parietal endoderm), regardless of the genotype of the injected cells:

Rebuttal Figure 15. Left: UMAP dimensionality reduction plots of wildtype embryo data taken from (Pijuan-Sala et al., 2019), highlighting extra-embryonic celltypes. Middle: mapping our cells onto this reference dataset, shown in red are cells which are the nearest neighbour in the reference data to cells from our data. Top panel shows WT host cells; middle shows Tet-KO chimeric cells; bottom shows WT chimeric cells. Right: barplots display the numbers of each celltype for WT host cells (top), Tet-KO chimeric cells (middle) and WT chimeric cells (bottom).

To clarify this, we have updated the text as follows:

“As expected from chimaeras generated from ESC injection into blastocysts, we find no contribution of tdTomato+ cells in the trophoblast compartment (ExE ectoderm cells), and this is true of WT and TKO cells.”

9. In line 217, for the scNMTseq how many cells were used? (only cells that passed the test are shown currently).

We have updated the text as suggested to include these figures (reproduced below). In addition we now include Supplementary Figure 11 which displays quality control metrics for these cells and the cutoffs used to filter low quality cells.

“In total we sequenced 768 cells using our multi-omic technology together with an additional 1056 cells using only scRNA-seq. The increased sample size of scRNA-seq data was used to aid cell type mapping. 1634, 724 and 616 cells passed quality control thresholds for RNA expression, DNA methylation and chromatin accessibility, respectively.”

10. In Figure S5A two WT cells contribute to ExE ectoderm (is there some variability between the embryos?)

The two samples with contributions to the ExE ectoderm are the WT tdTomato- host cells. These are not derived from ESCs and are expected to contribute to the formation of ExE ectoderm.

Rebuttal Figure 16. Left: Mapping WT tdTomato- (host WT cells) to the reference atlas. Shown are UMAP plots from (Pijuan-Sala et al., 2019) with red dots indicating nearest neighbours to cells in our wildtype host dataset. Right: bar plots displaying numbers of each cell type for the same embryos as left.

In contrast, the tdTomato+ injected ESCs (WT and Tet-TKO), do not contribute to the formation of ExE ectoderm:

Rebuttal Figure 17. Left: Mapping WT tdTomato- (host WT cells) to the reference atlas. Shown are UMAP plots from (Pijuan-Sala et al., 2019) with red dots indicating nearest neighbours to cells in our wildtype host dataset. Right: bar plots displaying numbers of each cell type for the same embryos as left.

As discussed above, this is exactly as expected, since ESCs introduced in a chimaera assay are expected to contribute only to the embryonic fraction and therefore act as a control for the experiment and the cell type mapping algorithm are performing as expected.

11. Throughout manuscript use "active DNA demethylation" instead of "active demethylation".

The text has been updated accordingly - and in most cases, 'active' is now replaced with 'TET-dependent' per comment 13.

12. Use "Dnmt and Tet mutant mouse embryos" instead of "a variety of DNA methylation mouse embryos"

The text has been updated accordingly.

13. In Line 39, and few other places, it is presumed that Tet mediated DNA demethylation leads to active demethylation only. Tet-mediated hydroxylation can drive passive demethylation by Uhrf1 evasion. This should be corrected in the manuscript.

The reviewer is correct. Our intention was to analyse TET dependent (vs. TET independent) demethylation rather than active vs. passive. We have updated the text to clarify this both in the introduction:

“Removal of CpG methylation from the genome can be achieved by passive dilution, in which DNMT1 is prevented from copying methylation onto daughter strands during replication and this is the major contributor to global demethylation events(von Meyenn et al., 2016). **De-methylation can also occur via enzymatic oxidation of methyl-cytosine into hydroxymethyl-cytosine and other oxidised derivatives catalysed by the Ten-eleven-translocation (TET) family of enzymes**(He et al., 2011; Ito et al., 2011; Tahiliani et al., 2009). **These oxidised bases can be removed and replaced by unmodified cytosine by base excision repair**(He et al. 2011; Maiti and Drohat 2011; Weber et al. 2016), **or can lead to replicative dilution due to Uhrf1 evasion**(Hashimoto et al. 2012; Otani et al. 2013).”

And in the section describing the scNMT-seq results:

“Impaired erythropoiesis in Tet-TKO cells is linked to lack of TET-dependent DNA demethylation of enhancer elements

We next sought to explore how impaired demethylation might be driving the failure to form primitive blood cells in Tet-TKO embryos. To our knowledge, DNA methylation has never been profiled during primitive erythropoiesis. However, previous studies have reported global loss of DNA methylation during definitive erythropoiesis (Shearstone et al., 2011). The decreased expression of DNMTs along this trajectory, and the requirement for DNA replication (Shearstone et al., 2011) suggested that this phenomenon is driven by passive DNA demethylation. However, given the phenotype we observe in Tet-TKO embryos, we hypothesised an involvement of the **TET-dependent** DNA demethylation pathway.”

14. Please include reference for Tet TKO ESCs in the methods section (it is noted in the results but not in methods)

This citation has been added to the methods section.

Reviewer #3:

Summary

In this manuscript, the authors present single-cell RNA-seq data from *Dnmt1*^{-/-}, *Dnmt3a*^{-/-}, *Dnmt3b*^{-/-}, and Tet-TKO in mouse E8.5 embryos, systematically explored the transcriptomic effect of DNA methylation writer and eraser perturbations in mouse early organogenesis. *Dnmt3a*^{-/-} and *Dnmt3b*^{-/-} show minor transcriptomic impact on E8.5, while *Dnmt1*^{-/-} shows developmental delay and over-expression of pluripotency and extra-embryonic genes. Tet-TKO shows biased lineage with the largest reduction in Erythroid and Neural crest. To provide additional evidence on Tet-TKO impacting the erythropoiesis, the authors also used their multi-omic technology, scNMT-seq, to further profile haemato-endothelial trajectory cells from Tet-TKO embryos. The authors found that TET-dependent demethylation in lineage enhancers is independent of chromatin accessibility changes, suggesting that active demethylation is required in erythropoiesis. Overall, this manuscript comprehensively discusses the DNA methylation regulator functions during early organogenesis at the cell-type-specific level. The analysis is well organized and easy to follow, and the findings are intriguing. I have several comments for the authors to improve their manuscript.

Major Comments:

The authors should provide more details on the scNMT-seq dataset since the title emphasizes that this manuscript is "multi-omic" profiling:

1) what are the general statistics and quality control metrics (coverage, number of genes detected, overall GpC, CpG level, bisulfite non-conversion rate, etc) for the scNMT-seq cells?
A supplementary figure (like Fig. S1 for scRNA-seq) is missing.

We thank the reviewer for pointing this out. Indeed we missed a quality control figure for the scNMT-seq cells. We have incorporated this as **Supplementary Figure 11**. (reproduced below),

Rebuttal Figure 18 (Figure S11): Quality control (QC) metrics for scNMT-seq Tet-TKO embryos.

(a) scRNA-seq QC metrics. Left: histograms showing the distributions of (i) number of detected genes per cell (ii) percentage of reads mapping to the mitochondrial chromosome per cell (iii) percentage of reads mapping to ribosomal genes per cell. Right: boxplots showing the same statistics as left but shown separately for each sample.

(b) Methylation QC metrics. Left: scatter plot comparing the global CpG methylation rate (x-axis, i.e. mean methylation across all CpGs in a given cell) to the CpG coverage per cell (y-axis). High quality cells are expected to have a large number of observed CpGs and the global rate to be $\geq 50\%$. Right: boxplots showing the same statistics as left but shown separately for each sample.

(c) Accessibility QC metrics. Left: scatter plot comparing the global GpC accessibility rate (x-axis, i.e. mean accessibility across all GpCs in a given cell) to the GpC coverage per cell (y-axis). High quality cells are expected to have a large number of observed GpCs and the global rate to be $\geq 10\%$ and $\geq 40\%$. Right: boxplots showing the same statistics as left but shown separately for each sample.

(d) Number of cells that pass QC for each data modality. Note that we sequenced 768 cells using scNMT-seq (three data modalities) together with an additional 1056 cells using only scRNA-seq. The increased sample size of scRNA-seq data was used to aid cell type annotation. A total of $N=562$ cells passed QC for all three data modalities.

(e) Bisulfite conversion rates (%) for each sample. Each dot corresponds to an individual cell.

2) Does the author have different QC criteria for each modality of the scNMT-seq? The method does not explain how the cell numbers on Line 216 are determined.

We have updated the methods section to include the QC criteria used (reproduced below). Additionally, the cutoffs for each metric used in filtering are indicated in the supplementary figure above.

“Low quality cells were excluded based on cytosine coverage (< 5,000 CpGs for methylation data and <10,000 GpCs for accessibility data) and global methylation (greater than 50%) and accessibility (between 10% and 40%) values.”

3) When mapping the scNMT-seq cell to the RNA reference atlas from Ref 21, does the author only use transcriptome information? Are there cells passing Met or NOME QC but failed RNA QC? Will these cells be included?

The reviewer is correct, we only use the transcriptome modality of the scNMT-seq cells for cell type mapping (since the reference dataset is scRNA-seq only). This means that any cells which fail QC for RNA but pass QC for the epigenome measurements will not be used in the downstream analysis. This amounts to 48 cells for DNA methylation and 34 cells for chromatin accessibility.

4) Fig 4e, f seems to rely on an unpublished scATAC-seq study (line 234) to determine distal lineage-specific regulatory elements. Is it possible to *de novo* identify potential regulatory elements with the current scNMT-seq data in this study? Such as performing differentially methylated region via CpG methylation or differentially accessible region via GpC methylation? It is hard to evaluate an unpublished dataset, yet the regions from that study are critical to the main finding of this manuscript.

We apologise for this omission in our first submission - the unpublished manuscript is now available on BioRxiv (<https://doi.org/10.1101/2022.06.15.496239>) and the data is available on GEO (<https://www.ncbi.nlm.nih.gov/geo/query/acc.cgi?acc=GSE205117>, reviewer token: gzcxaugylriplkn). Briefly, we created a transcriptomic and epigenetic atlas of mouse early organogenesis by performing scRNA-seq and scATAC-seq from the same cell. We used the combination of RNA expression and chromatin accessibility data to define a catalogue of celltype-specific regulatory elements which we employed in this manuscript.

Regarding the reviewer’s suggestion to derive regulatory regions from the scNMT-seq data. Indeed, this is theoretically possible, and is something we have attempted with previous datasets. However, the limited number of cells in scNMT-seq and its background signal makes it challenging for the *de novo* identification of regulatory elements (Nordström et al., 2019). When having high-quality ChIP-seq or ATAC-seq data from the same biological system (either bulk or single-cell), we recommend using these technologies that are optimal for *de novo* peak calling. In addition, the use of orthogonal data sets to generate these annotations makes the analysis less likely to suffer from technology-specific biases (Nordström et al., 2019).

To illustrate the challenges of performing *de novo* feature identification with scNMT-seq, we attach below a genome browser snapshot that overlays the scNMT-seq coverage with the scATAC-seq

coverage from matching cell types. While there is broad agreement in open chromatin regions, the lower background levels and the higher number of cells in scATAC-seq makes peak calling easier.

Rebuttal Figure 19. Genome browser plot showing scATAC-seq (dark blue) and scNMT-seq (light blue) accessibility coverage around hemoglobin beta loci for three matching cell types. The scATAC-seq coverage is extracted from (Argelaguet et al. 2022). Note that the number of cells for scATAC-seq is much higher than in scNMT-seq.

5) For haematoendothelial enhancer regions shown in Fig. 4e, are these enhancer regions intergenic or intragenic? Do they locate near the promoter or gene body of DEGs in Figure 3? Can the authors further associate some of these enhancers with DEGs in Figure 3?

The cell type specific enhancers are defined using the reference scRNA-seq/scATAC-seq atlas of mouse early organogenesis. In particular, we focus on distal open chromatin regions (i.e. ATAC peaks that do not overlap with transcription start sites). An example of a genome browser snapshot of the ATAC signal that we use to define cell type enhancers is shown below:

Rebuttal Figure 20. Genome browser plot of the *Gata6* locus. Each data track shows chromatin accessibility using scATAC-seq pseudobulked per celltype.

We briefly describe the procedure we used to define cell type-specific marker peaks: first, we performed pairwise differential accessibility analysis at the level of ATAC peaks. Then, for each cell type, we labelled as marker peaks those hits that are differentially accessible and upregulated in the cell type of interest in more than 85% of the comparisons. The bar plot below shows the number of marker peaks for each of the cell types that define the haematoendothelial trajectory (left plot). As mentioned, we focus on distal regulatory regions by excluding ATAC peaks that overlap with transcription start sites. The pie plot shows that, on average, ~40% of marker peaks are located in intergenic regions and ~55% within intragenic regions (introns and exons). All these are categorised as putative enhancers.

Rebuttal Figure 21. Defining cis-regulatory elements (putative enhancers) that mark the cell types that underlie the erythropoiesis trajectory.

- (a) UMAP of the reference data set, highlighting the three major cell types that define the erythropoiesis trajectory (Haematoendothelial progenitors, Blood progenitors and Erythroids). Note that in this study we merged the cell types “Blood progenitors 1/2” into “Blood progenitors”, “Erythroid 1/2/3” into “Erythroids”. This was done to obtain enough cells per cell type in the scNMT-seq analysis.
- (b) Bar plots displaying the number of putative enhancers that mark each cell type.
- (c) Classification of putative enhancers depending on their genomic context.

Next, as suggested by the reviewer, we linked marker peaks to genes that are differentially expressed (DE) between WT and *Tet*-TKO cells in at least one cell type of the haematoendothelial trajectory. The links were done based on a maximum genomic distance of 50,000kb from the centre of the ATAC peak to the TSS, which is a conservative estimate to prevent false positive associations. We find that ~51% of marker peaks can be linked to at least one DE gene, and 42% of DE genes can be linked to at least one marker peak.

A few examples of links between regulatory regions and DE are shown below:

Rebuttal Figure 22. Examples of individual cis-regulatory regions that are dysregulated in Tet-TKO cells.

(a) Genome browser plots of loci containing differentially expressed genes between WT and Tet-TKO cells at hematopoietic cell types. Shown is the ATAC-seq coverage across cell types from the reference data set and the DNA methylation coverage in the scNMT-seq data set, grouped by WT (blue) and KO (red). Highlighted with dashed lines are cis-regulatory regions identified from the ATAC-seq that are differentially methylated between WT and Tet-TKO cells and are linked to genes that are differentially expressed between WT and Tet-TKO cells at hematopoietic cell types.

(b) RNA expression values of the same genes as in (a), grouped by cell type.

Minor Comments

The sorting strategy looks the same between Fig. 3a and Fig. 4a schematics, but as mentioned in the method (line 378), scNMT sorting is labeled with additional antibodies to enrich specific cell populations. Fig. 4a should reflect that difference.

We thank the reviewer for this suggestion, which was also raised by reviewer 1. We have updated Figure 4a accordingly (reproduced below).

Rebuttal Figure 23 (and Figure 4a in the revised MS). Schematic summarising the scNMT-seq chimaera assay. Fluorescently labelled Tet-TKO ESCs are injected into wildtype blastocysts, transferred into pseudopregnant hosts then collected at E8.5. FACS is used to isolate specific populations (CD41+, erythroid; KDR+, Haematoendothelial progenitors; CD41+ KDR+, blood progenitors and CD41-, KDR-) of both labelled KO cells (red) and non-labelled WT host cells (blue) which are processed and sequenced using scNMT-seq.

Line 174, there is no Figure S5f.

We thank the reviewer for pointing this out. We have now amended this.

How is the PAGA graph generated in Fig. 1b and Fig. S8a? Not mentioned in the method section.

In our first submission we used the Partition-based graph abstraction method (PAGA) implemented in scanpy (Wolf et al., 2019) to provide an interpretable graph-like map of the reference UMAP that could be used to visualise gene expression at the cell type resolution. Although we used this data structure in the exploratory analysis, we did not use it in any visualisation presented in the manuscript. Therefore we have decided to remove the PAGA graphs from the two figures where it was displayed.

How is the pseudo-time analysis done in Fig. 4? Not mentioned in the method section.

We have clarified this in the Methods section:

“The pseudotime order for the erythropoiesis trajectory was inferred using diffusion maps with the *destiny* R package (v3.8.1) (Angerer et al., 2016)”

Line 228, Gene *Uhrf1* is mentioned in the text but does not occur in Fig. 4c.

We thank the reviewer for pointing out this mistake. We have now incorporated *Uhrf1* in Figure 4c

Line 417, In the "Cell type annotation" part, how does the query cell be mapped to the reference UMAP after assigned cell types (all the reference cell UMAP coordinates were from Ref 21)?

This is a good question. In the Method section we have explained how the mapping between query and atlas cells is computed but not how the results are visualised using the UMAP from Ref 21.

First, we believe it is important to clarify how the mapping is calculated: we infer joint low-dimensional space that includes both query and atlas cells. In this low-dimensional space (which is a PCA space with 50 dimensions in our case) we perform batch correction between query and atlas and subsequently use the batch-corrected PCA space to compute a k-nearest neighbours (kNN) graph that links each query cell to the closest atlas cells. We use this kNN graph to transfer cell type assignments from the atlas to the query cells.

To visualise the results of the mapping one option could be to generate a UMAP of the joint kNN that includes both query and atlas cells, as shown below:

Rebuttal Figure 24. UMAP dimensionality reduction plot of scRNA-seq using both atlas data and our (query) data. In the left cells are coloured by query or atlas, in the right cells are coloured by cell type assignment

The problem with this approach is that UMAP is a stochastic algorithm that yields different results at each run. When using reference data sets one would like to preserve the precomputed UMAP representations, in this case the one originally published in (Pijuan-Sala et al. 2019), as this facilitates data interpretation. To achieve this, we plotted the reference UMAP (with no query cells) and used the joint kNN graph to highlight the cells that are nearest neighbours to the query cells.

We hope that this clarifies the reviewer's question. We have added a sentence in the Methods section to better explain how the visualisation of the mapping is done.

I have a concern about data access. Currently, all the links in the data availability section are not available to visit. In addition, a GEO accession # (or another public database accession) is also not provided.

We apologise that the data was not accessible in our previous submission. The data can now be found here <https://www.ncbi.nlm.nih.gov/geo/query/acc.cgi?acc=GSE204908>.

The “Data availability” section has now been updated to include the GEO accession number. Links to processed data objects as well as to R Shiny apps for interactive data analysis are available in the corresponding github repositories displayed in the “Code availability” section.

Bibliography

Angerer, P., Haghverdi, L., Büttner, M., Theis, F.J., Marr, C., and Buettner, F. (2016). destiny: diffusion maps for large-scale single-cell data in R. *Bioinformatics* 32, 1241–1243. .

Argelaguet, R., Lohoff, T., Li, J.G., Nakhuda, A., Drage, D., Krueger, F., Velten, L., Clark, S.J., and Reik, W. (2022). Decoding gene regulation in the mouse embryo using single-cell multi-omics.

Dahlet, T., Argüeso Lleida, A., Al Adhami, H., Dumas, M., Bender, A., Ngondo, R.P., Tanguy, M., Vallet, J., Auclair, G., Bardet, A.F., et al. (2020). Genome-wide analysis in the mouse embryo reveals the importance of DNA methylation for transcription integrity. *Nat. Commun.* 11, 3153.

Grosswendt, S., Kretzmer, H., Smith, Z.D., Kumar, A.S., Hetzel, S., Wittler, L., Klages, S., Timmermann, B., Mukherji, S., and Meissner, A. (2020). Epigenetic regulator function through mouse gastrulation. *Nature* 584, 102–108.

He, Y.-F., Li, B.-Z., Li, Z., Liu, P., Wang, Y., Tang, Q., Ding, J., Jia, Y., Chen, Z., Li, L., et al. (2011). Tet-mediated formation of 5-carboxylcytosine and its excision by TDG in mammalian DNA. *Science* 333, 1303–1307.

Ito, S., Shen, L., Dai, Q., Wu, S.C., Collins, L.B., Swenberg, J.A., He, C., and Zhang, Y. (2011). Tet proteins can convert 5-methylcytosine to 5-formylcytosine and 5-carboxylcytosine. *Science* 333, 1300–1303.

Kinoshita, M., Li, M.A., Barber, M., Mansfield, W., Dietmann, S., and Smith, A. (2021). Disabling de novo DNA methylation in embryonic stem cells allows an illegitimate fate trajectory. *Proc. Natl. Acad. Sci. U. S. A.* 118. <https://doi.org/10.1073/pnas.2109475118>.

von Meyenn, F., Iurlaro, M., Habibi, E., Liu, N.Q., Salehzadeh-Yazdi, A., Santos, F., Petrini, E., Milagre, I., Yu, M., Xie, Z., et al. (2016). Impairment of DNA Methylation Maintenance Is the Main Cause of Global Demethylation in Naive Embryonic Stem Cells. *Mol. Cell* 62, 848–861.

Ng, R.K., Dean, W., Dawson, C., Lucifero, D., Madeja, Z., Reik, W., and Hemberger, M. (2008). Epigenetic restriction of embryonic cell lineage fate by methylation of E1f5. *Nat. Cell Biol.* 10, 1280–1290.

Nordström, K.J.V., Schmidt, F., Gasparoni, N., Salhab, A., Gasparoni, G., Kattler, K., Müller, F., Ebert, P., Costa, I.G., Pfeifer, N., et al. (2019). Unique and assay specific features of NOME-, ATAC- and DNase I-seq data. *Nucleic Acids Research* 47, 10580–10596. <https://doi.org/10.1093/nar/gkz799>.

Pijuan-Sala, B., Griffiths, J.A., Guibentif, C., Hiscock, T.W., Jawaid, W., Calero-Nieto, F.J., Mulas, C., Ibarra-Soria, X., Tyser, R.C.V., Ho, D.L.L., et al. (2019). A single-cell molecular map of mouse gastrulation and early organogenesis. *Nature* 566, 490–495.

Sakaue, M., Ohta, H., Kumaki, Y., Oda, M., Sakaide, Y., Matsuoka, C., Yamagiwa, A., Niwa, H., Wakayama, T., and Okano, M. (2010). DNA methylation is dispensable for the growth and survival of the extraembryonic lineages. *Curr. Biol.* 20, 1452–1457.

Shearstone, J.R., Pop, R., Bock, C., Boyle, P., Meissner, A., and Socolovsky, M. (2011). Global DNA demethylation during mouse erythropoiesis in vivo. *Science* 334, 799–802. .

Tahiliani, M., Koh, K.P., Shen, Y., Pastor, W.A., Bandukwala, H., Brudno, Y., Agarwal, S., Iyer, L.M., Liu, D.R., Aravind, L., et al. (2009). Conversion of 5-methylcytosine to 5-hydroxymethylcytosine in mammalian DNA by

MLL partner TET1. *Science* 324, 930–935.

Wolf, F.A., Hamey, F.K., Plass, M., Solana, J., Dahlin, J.S., Göttgens, B., Rajewsky, N., Simon, L., and Theis, F.J. (2019). PAGA: graph abstraction reconciles clustering with trajectory inference through a topology preserving map of single cells. *Genome Biol.* 20, 59.

Second round of review

Reviewer 1

The authors have addressed all my comments very satisfactorily and I am very happy with the revised manuscript. This study is timely, very interesting, and demonstrates the power of using single cell technologies to study development.

Reviewer 2

The authors have addressed all my minor concerns in detail. Manuscript is suitable for publication.

Reviewer 3

I have no further comments for improvement of this now excellent manuscript.